# Constitutively bound CTCF sites maintain 3D chromatin architecture and long-range epigenetically regulated domains

Amanda Khoury[1], Joanna Achinger-Kawecka [1], Saul A. Bert[1], Grady C. Smith[1], Hugh J. French [1], Phuc-Loi Luu[1], Timothy J. Peters [1], Qian Du[1], Aled J. Parry[1], Fatima Valdes-Mora [1], Phillippa C. Taberlay[1,2], Clare Stirzaker[1,3], Aaron L. Statham[1] & Susan J. Clark [1,3]*

The architectural protein CTCF is a mediator of chromatin conformation, but how CTCF binding to DNA is orchestrated to maintain long-range gene expression is poorly understood. Here we perform RNAi knockdown to reduce CTCF levels and reveal a shared subset of CTCF-bound sites are robustly resistant to protein depletion. The 'persistent' CTCF sites are enriched at domain boundaries and chromatin loops constitutive to all cell types. CRISPR-Cas9 deletion of 2 persistent CTCF sites at the boundary between a long-range epigenetically active (LREA) and silenced (LRES) region, within the Kallikrein (*KLK*) locus, results in con-cordant activation of all 8 *KLK* genes within the LRES region. CTCF genome-wide depletion results in alteration in Topologically Associating Domain (TAD) structure, including the merging of TADs, whereas TAD boundaries are not altered where persistent sites are maintained. We propose that the subset of essential CTCF sites are involved in cell-type constitutive, higher order chromatin architecture.

[1] Epigenetics Research, Genomics & Epigenetics Division, Garvan Institute of Medical Research, Sydney, NSW 2010, Australia. [2] School of Medicine, University of Tasmania, Hobart, TAS 7000, Australia. [3] St Vincent's Clinical School, UNSW Sydney, Sydney, NSW 2000, Australia. *email: s.clark@garvan.org.au

Three-dimensional (3D) chromatin conformation is an important regulator of gene expression as it affects which regulatory elements come into contact with gene promoters, and thus which genes are activated and which are repressed[1]. Hi-C allows simultaneous capture of all chromatin interactions occurring across the genome in a single experiment and has revealed that chromatin interactions are compartmentalised into topologically associated domains (TADs)[2,3]. TADs are ~1 Mb sized, contiguous chromosomal regions with insulated boundaries. This organisation facilitates a high frequency of interaction for loci located within a TAD and little interaction between TADs[1,2]. TADs are reportedly conserved across cell types[2,4,5]. A vast amount of research has established CCCTC-binding factor (CTCF) as a mediator of chromatin looping[6,7] and TAD boundary insulation[8,9]. In addition to these mechanisms, CTCF can also insulate between regions of active chromatin marked by H3K4me3 and repressive regions marked by H3K27me3 to prevent aberrant spreading of either chromatin mark into its opposing state[10,11].

However, several studies have yielded inconsistent results on the role of CTCF in long-range chromatin interactions and in the maintenance of TAD boundaries. For example it has been reported that CRISPR-Cas9 depletion of CTCF located at candidate TAD boundaries is sufficient to deplete the targeted boundary[8,11] and conversely, that TAD boundaries remain intact following loss of CTCF[12]. Moreover, a CTCF siRNA approach in HEK293T cells reported a general maintenance of TAD boundaries and modest changes to gene expression[13], whereas, a more recent study that utilised the auxin-inducible degron (AID) system to directly target CTCF protein for degradation (as opposed to targeting mRNA) in mouse embryonic stem cells observed complete loss of insulation at 80% of TAD boundaries[14]. Taken together, these studies demonstrate that the consequences of CTCF depletion on 3D architecture are still unclear.

We previously reported long-range epigenetic silencing (LRES)[15] and activation (LREA)[16] in prostate cancer. In LRES, regions of chromatin concordantly lose active histone marks and gain repressive histone marks in cancer cells, which results in the underlying genes becoming silenced. In contrast to this, in LREA, regions of chromatin concordantly lose repressive histone marks and gain active histone marks in cancer cells and undergo gene activation. One long-range deregulated region of interest in prostate cancer is located on chr19—51,322,404–51,587,502 (hg19). This locus is ~270 kb in size and contains all members of the Kallikrein (KLK) serine-protease gene family, including KLK3, which encodes prostate-specific antigen (PSA),[17,18] and KLK4, which is implicated as a mediator of mTOR signalling in prostate cancer[19]. Intriguingly, KLK3 and KLK4, as well as KLK1, KLK15, KLK2, KLKP1 are contained within a LREA region in prostate cancer cells, which is immediately adjacent to an LRES region harbouring KLK5, KLK6, KLK7, KLK8, KLK9, KLK10, KLK11, KLK12, KLK13 and KLK14 genes. We were therefore motivated to determine if CTCF sites bounded the border of the LREA and LRES regions at this locus.

Here, we explore the genome-wide chromatin effects of global CTCF depletion and CRISPR-targeted CTCF deletion to determine the involvement of CTCF in compartmentalisation of the long-range epigenetically regulated regions. We show that there is a subset of CTCF sites, that are resistant to CTCF depletion and propose these persistent CTCF sites are essential for cell-type constitutive higher order chromatin architecture and the maintenance of long-range epigenetically regulated domains.

## Results

### The Kallikrein locus is bordered by CTCF-binding sites. The Kallikrein (KLK) gene locus is comprised of an LREA region,

immediately adjacent to a LRES region (Fig. 1). Given the discrete compartmentalisation of gene expression at the KLK locus and the well-established role of CTCF as an insulator of functional domains[11,20,21], we were interested to determine if CTCF binding was associated with the demarcation of the active and repressive domains. We analysed CTCF ChIP-seq data for normal prostate cells (PrEC) and prostate cancer cell line (LNCaP)[16] and found that the CTCF-binding pattern was strikingly similar across the locus regardless of the different expression profiles. Both normal and cancer cells harboured two discrete CTCF-binding sites at the boundary of the active and repressive regions, as well as CTCF sites throughout the flanking domains (Fig. 1).

To next evaluate the chromatin structure at this locus we performed Chromosome Conformation Capture (3C)[22] using a fragment containing one of the two CTCF sites at the boundary of the active LREA and repressive LRES regions, indicated in Fig. 2a, as bait. We found two interactions; one occurred 149.2 kb upstream of the bait and the other 163.8 kb downstream from the bait (Fig. 2a, Supplementary Fig. 1a). These interactions were also verified by performing reciprocal 3C using the long-range flanking interacting fragments as baits (Fig. 2b, c; Supplementary Fig. 1b, c). However in comparison to the strong interaction between the fragments located at 149.2 and 163.8 kb from the LRES/LREA boundary there is relatively little interaction between the outer borders of the KLK domains. Together this data demonstrates that discrete chromatin loops spatially separate the LREA and LRES regions across the KLK gene locus at the boundary of CTCF sites. CTCF-binding motifs flanking chromatin loops mostly have a convergent orientation[4,23]. The downstream loop at the KLK locus has anchors with convergent CTCF motifs, however the upstream loop is potentially anchored by divergent CTCF motifs (Fig. 1).

**CTCF depletion does not lead to widespread gene activation.** As CTCF has been reported to act as a barrier to the spreading of active and repressive marks[11,24,25], we assessed whether loss of CTCF results in gene expression and chromatin modification alterations both genome-wide and at the KLK gene locus. We depleted CTCF in LNCaP cells with a siRNA pool (4 siRNAs/pool) and showed robust depletion (≥90%) of CTCF mRNA by qRT-PCR (Fig. 3a) and protein by western blot (Fig. 3b) at both 72 and 144 h post-transfection. We first measured global changes to gene expression at 24, 72 and 144 h post CTCF-siRNA transfection using an Affymetrix GeneChip® Human Gene 2.0 ST Array. After 24 h only CTCF mRNA was significantly down regulated. At 72 and 144 h there were 17 and 663 genes (1.9% of total on array) that showed altered expression, respectively (Fig. 3c). We found that 15/17 genes at 72 h had CTCF bound at their promoter prior to transfection and of these, 15/15 lost CTCF binding following RNAi treatment. At 144 h, 155/663 had CTCF bound at their promoters prior to transfection and 145/155 lost binding following RNAi. However, there was no significant change to gene expression across the KLK LREA/LRES regions (Fig. 3d). We performed 3C following CTCF knockdown to assess if there were any changes to looping at the KLK locus. Loss of CTCF did not lead to the formation of new contacts in the region. However, on average we saw 16.9% and 36.3% reduction in interaction strength between the bait fragment and its upstream and downstream targets, respectively (Supplementary Fig. 2) suggesting that the lost CTCF sites are less critical for the loop formation and gene expression control at the KLK locus.

We next performed H3K4me3 and H3K27ac ChIP-seq following 144 h of CTCF RNAi treatment. We found that there was no significant change to the active histone marks across the KLK LREA/LRES regions (Fig. 3e), and only a small proportion of

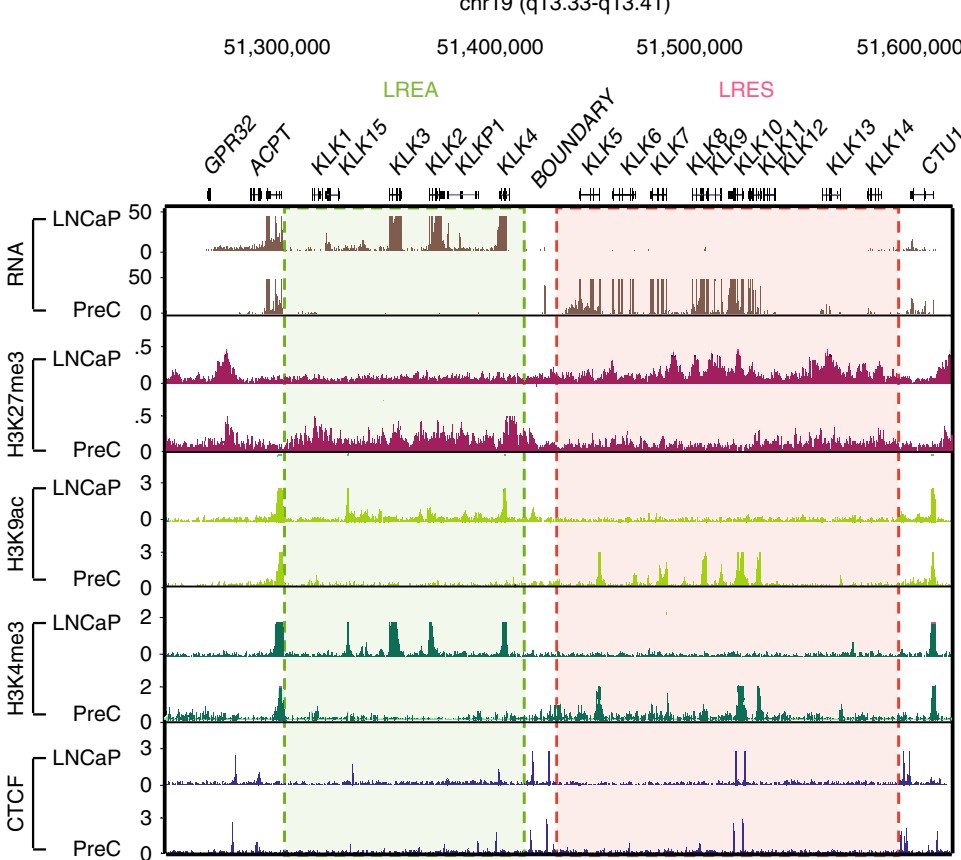

**Fig. 1 Epigenome map of the Kallikrein locus in prostate cells.** RNA-seq and ChIP-seq (H3K27me3, H3K9ac, H3K4me3 and CTCF) profiles of adjacent, transcriptionally active (green highlight) and silenced (red highlight) regions in LNCaP cancer cell line and normal PrEC cells. CTCF ChIP-seq shows CTCF binding throughout the locus. Two CTCF sites are located at the boundary between the active and silenced regions.

H3K27ac peaks (1773/55,700) and H3K4me3 (193/12,037) peaks genome-wide were significantly altered in the control versus CTCF RNAi cells (Fig. 3f). Therefore, our results show that CTCF depletion does not lead to widespread ectopic gene activation or chromatin modification changes.

**CTCF-binding sites resistant to CTCF depletion.** Due to the modest molecular changes identified after CTCF knockdown we next performed CTCF ChIP-seq, following 144 h of CTCF RNAi treatment, to determine if the apparent robust mRNA and protein depletion of CTCF (Fig. 3a, b) was sufficient to result in direct loss of CTCF binding to chromatin genome-wide. Intriguingly, we found that out of the 25,617 CTCF ChIP-seq bound sites, ~11.6% (2973/25,617) retained significant and persistent binding following treatment with CTCF siRNA in comparison to only 49 CTCF sites that were gained (Fig. 4a). 'Persistent' CTCF sites were defined as sites that were maintained across wild-type, control RNAi and the CTCF RNAi biological replicates. In contrast, 'lost' CTCF sites were defined as sites that were absent in the CTCF RNAi biological replicates, relative to the wild-type and control RNAi cells (Supplementary Fig. 3a). Example loci that demonstrate both lost and persistent CTCF sites are shown (Fig. 4b and Supplementary Fig. 3b). Notably, we found that the majority of the CTCF sites were lost across the *KLK* gene locus in LNCaP cells following RNAi treatment, with the exception of two persistent CTCF sites located at the border of the LREA/LRES regions (Fig. 4c), which we confirmed were maintained by ChIP-qPCR (Supplementary Fig. 3c). Since the cohesin complex has

been established to co-bind at CTCF sites in a CTCF-dependent manner[26], we also assessed if cohesin binding was associated with lost or persistent CTCF sites. We performed ChIP-qPCR for the cohesin subunit, RAD21, at the *KLK* locus following 144 h of CTCF RNAi. Our results demonstrate that RAD21 binding was reduced where CTCF binding was lost and was retained at the two internal border persistent CTCF sites (two-tailed *t*-test, $p < 0.05$) (Fig. 4c).

**CRISPR of persistent CTCF-binding sites at *KLK* locus.** To investigate if the two persistent boundary CTCF sites at the *KLK* locus were important in the maintenance of the functional demarcation of the LREA and LRES regions, we employed CRISPR-Cas9n[27] to delete these CTCF-binding motifs (location of guide RNAs are featured in Supplementary Fig. 4). We achieved an 80.2% depletion of CTCF binding at the upstream motif (two-tailed *t*-test, $p < 0.0001$) and 47.8% depletion at the downstream motif (two-tailed *t*-test, $p = 0.0002$) as measured by CTCF ChIP-qPCR (Fig. 5a). Importantly, binding at the remaining CTCF motifs in the *KLK* region was unchanged. We also performed a RAD21 ChIP-qPCR and found that RAD21 binding was lost at the two CRISPR sites, but retained at all other CTCF sites across the region (Fig. 5b). We next assessed if there were expression changes to the *KLK* genes within the LREA and LRES regions following CRISPR-Cas9n using qRT-PCR. We found no change in gene expression within the LREA loop, whereas there was coordinate up regulation of all the *KLK* genes within the LRES region (Fig. 5c and Supplementary Fig. 5).

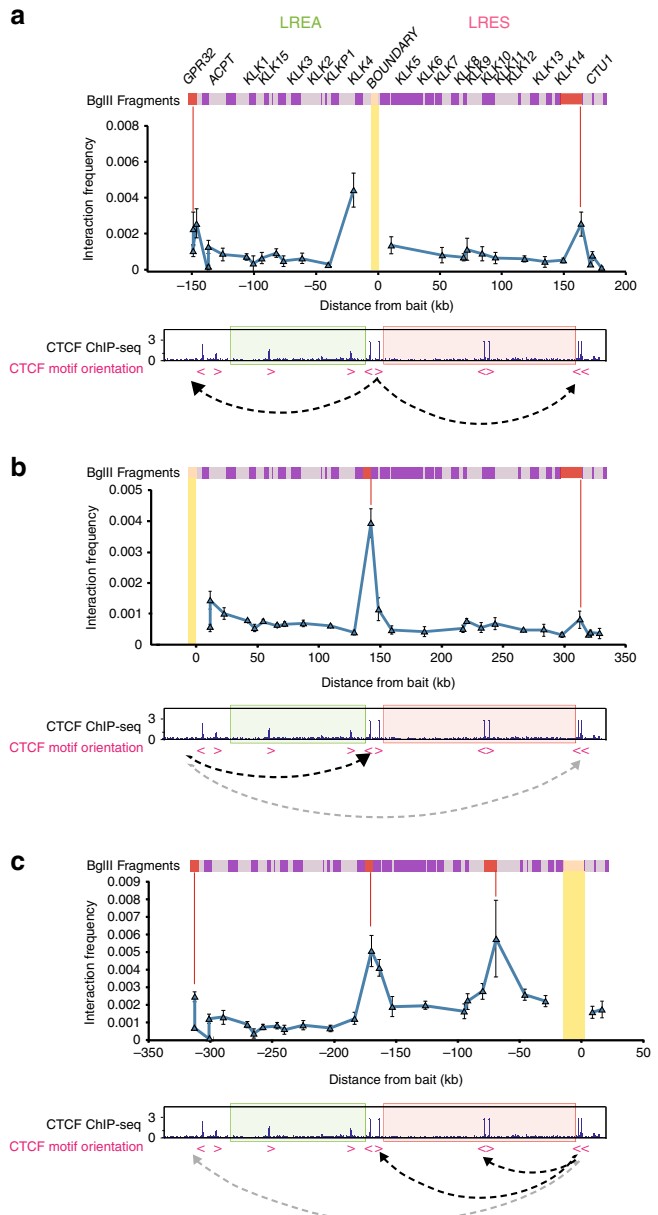

**Fig. 2 3D structure of the *KLK* locus compartmentalises expression domains. a** 3C-qPCR at the *KLK* locus. Bars above the graph illustrate BglII fragments across the region. The 3C bait fragment is indicated by the yellow bar and interacting fragments are shown by red lines (149.2 kb upstream from bait, 163.8 kb downstream from bait). The pink arrows indicate the direction of CTCF motifs. The dotted arrows demonstrate the loops indicated by the interaction data. Error bars represent standard error (SE). **b** As in **a**, utilising upstream interacting fragment from **a** as bait. Interacting fragments are 143.2 and 313.0 kb downstream from the bait. **c** As in **a**, utilising downstream interacting fragment from **a** as bait. Interacting fragments are 68.9, 169.8 and 313.0 kb upstream from the bait.

To determine if the gene activation was accompanied by a change in chromatin conformation, we performed 3C on the CTCF–CRISPR–Cas9n and non-targeting CRISPR–Cas9n control cells in biological duplicate experiments (Fig. 5d and Supplementary Fig. 6a, b). On average a 66.1% reduction in chromatin interaction strength was observed, in the 3C replicates, between the bait and its downstream interacting fragment for the LRES loop containing the up-regulated genes with a lesser change to the LREA loop (12.6% reduction) (Fig. 5d and Supplementary

Fig. 6a, b). In addition there is no change in weak chromatin interactions observed at the outer borders of the LRES/LREA regions (Supplementary Fig. 6c). Taken together these results show that the removal of the region harbouring the persistent CTCF sites at the boundary of the LREA/LRES regions results in opening of the downstream chromatin loop and activation of the entire LRES domain, as summarised in the schematic in Fig. 5e.

**Persistent CTCF sites show stronger binding intensity**. To explore the characteristics of lost and persistent CTCF-binding sites genome-wide we first asked if persistent CTCF sites display more prominent CTCF binding prior to siRNA treatment, which could explain greater maintenance of CTCF binding-signal after knockdown. To address this we divided the CTCF ChIP-seq data for wild-type LNCaP cells into two categories: (1) lost following CTCF RNAi or (2) persistent following CTCF RNAi. We then plotted the average CTCF ChIP-seq signal at CTCF-binding sites, which revealed that on average persistent sites have stronger binding intensity than lost sites (two-tailed *t*-test, $p < 0.0001$) (Supplementary Fig. 7a). Given the nature of ChIP-seq data this result can also be interpreted as these sites being bound in more cells in the population. A heat map displaying the binding intensity at each CTCF site individually revealed that a proportion of lost sites also have a binding intensity equivalent to the persistent sites, which indicates that binding intensity is not the only factor contributing to CTCF stability after siRNA treatment (Supplementary Fig. 7b). It was previously reported that CTCF sites with the highest binding affinity have a CTCF-binding motif that is closer in sequence to the CTCF consensus motif[28]. We used HOMER to search for the frequency of CTCF consensus motifs in lost and persistent CTCF sites. In agreement with the literature this revealed that the consensus motif was identified in a greater proportion of persistent sites 2556/2949 (85.9%) than lost sites 13,475/22,644 (59.5%).

DNA methylation of the CTCF-binding motif has been shown to prevent the binding of CTCF[29,30]. Hence we asked whether there was a difference in the levels of average DNA methylation between the lost and persistent CTCF subsets. To assess the methylation status and DNA accessibility we analysed NOMe-seq data from wild-type LNCaP cells[31]. We divided the methylome data based on whether the CTCF sites had lost or persistent binding following CTCF RNAi. The CpG methylation values at each of the subclasses of CTCF sites revealed that persistent binding sites in general are less methylated than lost CTCF sites (Supplementary Fig. 8a). The 1-GpC methylation (measure of nucleosome occlusion) highlighted that the DNA immediately surrounding persistent CTCF sites are less occluded by nucleosomes and thus are more accessible for CTCF binding than the DNA surrounding the lost CTCF sites. To next compare persistent and lost CTCF peaks with similar high CTCF levels we analysed the subset of lost CTCF sites with the highest CTCF levels (2973) to an equal number of persistent sites. We found that the CpG methylation levels are equivalent at the strongly bound CTCF sites and accessibility is also similar between the two groups. However peak height at the CTCF-binding site is more pronounced for persistent sites, which suggests that these sites are more consistently bound across a population of cells (Supplementary Fig. 8b).

**Persistent CTCF sites are enriched at TAD boundaries**. To assess the effect of CTCF depletion on three-dimensional chromatin conformation across the genome we performed Hi-C experiments with 40 kb resolution in LNCaP cells following 144 h of transfection with control and CTCF siRNA. We first analysed gross changes to the TAD structure by determining the total

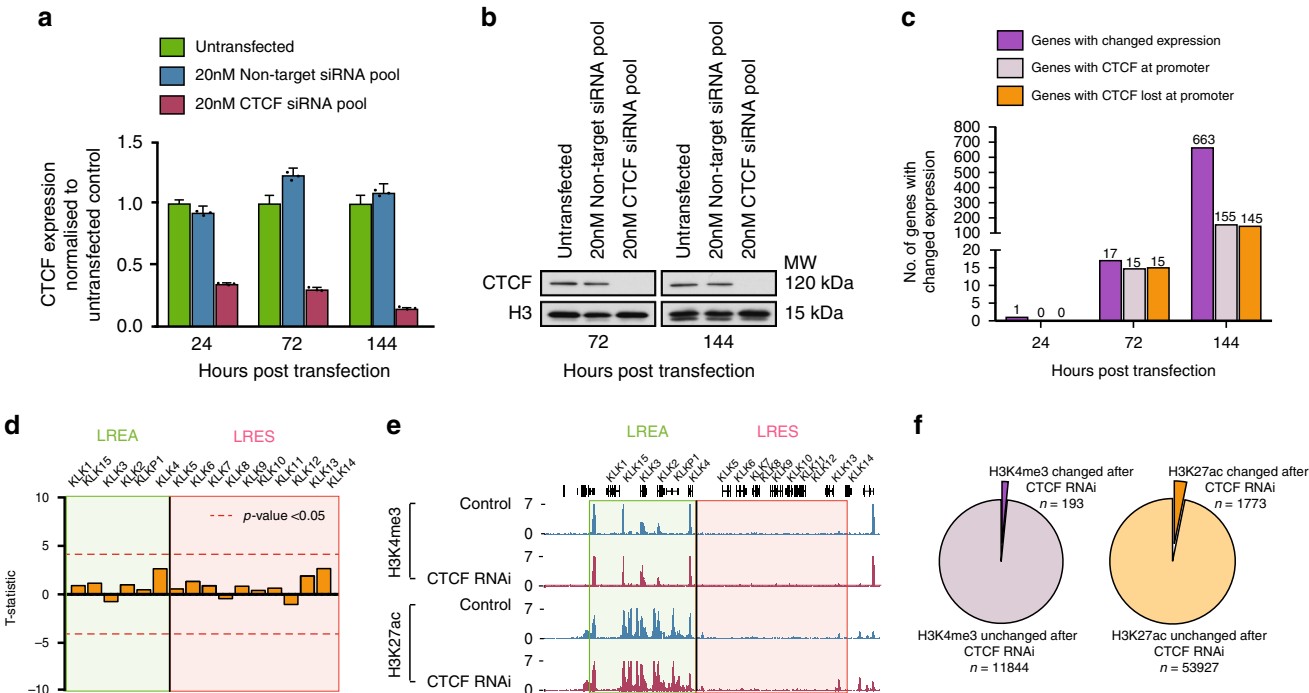

**Fig. 3 CTCF RNAi leads to modest changes to gene expression and chromatin. a** qPCR measuring relative levels of CTCF mRNA for untransfected, control siRNA and CTCF siRNA conditions at 24, 72 and 144 h following transfection of LNCaP cells with 20 nM control and CTCF siRNA. Error bars represent SE. Data for technical replicates $n = 3$ are overlaid as a dot blot. **b** Western blots for CTCF following 72 and 144 h of CTCF RNAi show >90% reduction of CTCF protein. Source data are provided as a Source Data file. **c** Bar graph shows total number of genes with changed expression following 24, 72 and 144 h of CTCF RNAi (dark purple). Light purple bars indicate the number of total genes with changed expression that have CTCF binding at their promoter. Orange bars show the number of this subset that loses CTCF binding following CTCF RNAi. **d** T-statistics show no significant difference ($p$-value < 0.05, indicated by dashed line) in *KLK* locus gene expression between control and CTCF RNAi samples (144 h post transfection) processed on an Affymetrix Gene Chip 2.0 expression array. $n = 2$ biologically independent experiments. **e** ChIP-seq for H3K4me3 and H3K27ac following 144 h of CTCF RNAi revealed no change to binding strength at *KLK* region. **f** Venn diagrams displaying proportion of H3K4me3 and H3K27ac modifications that undergo significant change in binding strength following 144 h of CTCF RNAi.

amount of the genome compartmentalised into TADs before and after CTCF siRNA knockdown. We found that a similar amount of the genome was packaged within TADs for the control and knockdown conditions (2.64 and 2.58 Gb, respectively). In the control condition, 2864 TADs were called with a median size of 720 kb. Interestingly the CTCF RNAi-treated cells contained less TADs (2389) with a larger median size of 840 kb (Supplementary Fig. 9).

To further explore the nature of TAD alterations following CTCF depletion we intersected TAD boundaries from the control and knockdown conditions and found that the majority of TAD boundaries, 2057/2609 (~79%) were maintained after CTCF siRNA treatment. Approximately 33% (1022/3079) were lost from the control cells and interestingly, 21% (552/2609) new TAD boundaries were identified in the CTCF siRNA knockdown cells (Fig. 6a). Moreover intersection of control and CTCF RNAi TADs revealed that 416 (14.5%) TADs in control cells became merged into larger TADs in the CTCF knockdown cells which is represented schematically in Fig. 6b. This is consistent with the smaller number of larger sized TADs suggesting that CTCF depletion results in merging of TADs due to loss of insulation. In comparison only 160 (6.7%) new smaller TADs were identified. Examples of merged TADs and subdivided TADs following CTCF siRNA knockdown are shown in Fig. 6c and d and Supplementary Fig. 10a, b. Notably maintained TAD boundaries after CTCF depletion overlapped with the location and maintenance of persistent CTCF-binding sites (example shown in Fig. 6c). A positional enrichment graph showing the distribution

of lost and persistent CTCF peaks at maintained TAD boundaries confirmed the genome-wide enrichment for persistent CTCF site binding at these loci (two-tailed $t$-test, $p < 0.0001$) (Fig. 6e). These results are consistent with a role for persistent CTCF sites in insulation of TAD boundaries from perturbation.

**Persistent CTCF sites and constitutive chromatin architecture**. Since CTCF binding can be cell-type specific, common to multiple cell types, or cell-type constitutive[32], we next asked if persistent sites were also conserved in other cell types and associated with chromatin architecture. We analysed kilobase resolution Hi-C data for eight diverse cell types—GM12878, K562, HeLA, IMR90, HUVEC, NHEK, HMEC and KBM7[4] and the corresponding CTCF ChIP-seq data sets for each of these cell types. Remarkably, we found that 97.7% of all the persistent CTCF sites were constitutively bound across all eight diverse cell types, in comparison to 67.5% of lost CTCF sites (Fig. 7a). Figure 7b shows an example of conservation of persistent sites across all cell types. To investigate if the persistent CTCF sites identified in the cancer LNCaP cells were also persistent in an unrelated cell type we performed CTCF knockdown in normal IMR90 cells and achieved ~80% knockdown of CTCF mRNA (Supplementary Fig. 11a). Notably we showed the majority of persistent CTCF sites in IMR90 cells (87.3%) overlapped with the LNCaP-persistent sites (Fig. 7c). Given this ubiquitous binding in human cell lines, we next used PhastCons[33] to determine the level of evolutionarily conservation of the CTCF-binding sites in the

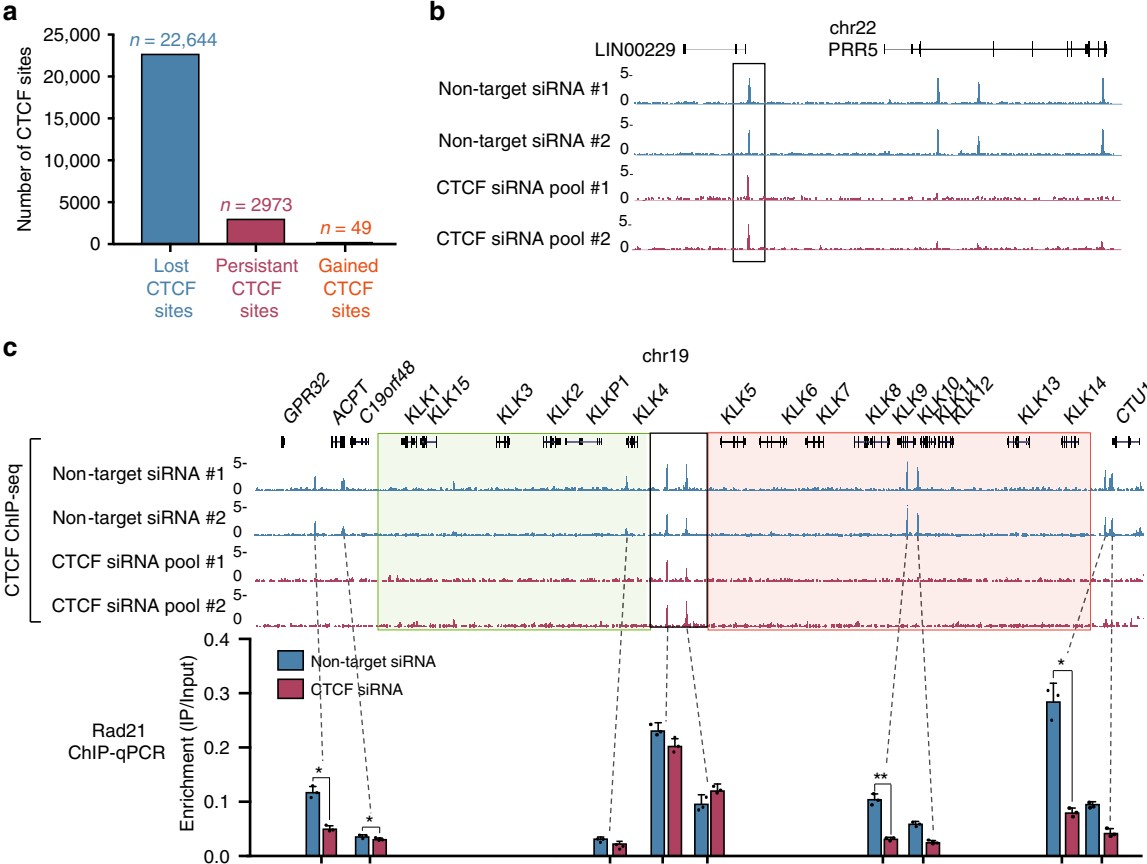

**Fig. 4 A subset of CTCF binding sites is more resistant to CTCF depletion. a** Bar plot quantifying 'lost', 'persistent' and 'gained' CTCF sites in LNCaP cells. **b** UCSC genome browser screenshot demonstrating maintenance of CTCF binding at some sites (box) following CTCF knockdown. **c** CTCF ChIP-seq illustrates CTCF binding at the boundary of the active and silenced regions at the *KLK* locus is retained following 144 h of CTCF RNAi. All other binding in the region is lost. RAD21 ChIP-qPCR performed after 144 h of control and CTCF RNAi demonstrates that cohesin binding is maintained where CTCF binding is persistent at the *KLK* locus (* indicates where two-tailed *t*-test, *p* < 0.05; *n* = 2 biologically independent experiments). Source data are provided as a Source Data file. Error bars represent SE. Data for technical replicates *n* = 3 are overlaid as a dot blot.

lost and persistent subsets. This analysis showed a much greater level of sequence conservation for the persistent CTCF-binding sites (Supplementary Fig. 11b). We also found a significant over-representation of persistent CTCF sites versus lost sites at domain boundaries in the eight diverse cell types. (Fig. 7d). Given the strong conservation of persistent sites, we next asked whether persistent sites were more enriched at cell-type constitutive domains. We divided all domain boundaries into: (1) cell-type specific (one cell type), (2) common to more than one cell type (2–7 seven cell types), or (3) constitutively present in all cell types (8 cell types) and created positional enrichment plots to overlay the distribution of LNCaP lost and persistent CTCF at the boundaries of each subset (Fig. 7e). We found that lost CTCF-binding sites were prevalent at cell-type-specific domains and the binding intensity decreased, as domains became more cell-type ubiquitous. In contrast, persistent CTCF site binding was deplete at cell-type-specific domains and increased in intensity at cell-type constitutive chromatin domains (two-tailed *t*-test, *p* = 0.02). We performed the same analysis on cell-type specific, cell-type common and cell-type constitutive chromatin loops and confirmed that persistent sites are also enriched at constitutive chromatin loops (Supplementary Fig. 11c).

TAD boundaries are enriched for housekeeping genes[2]. We therefore hypothesised that the cell-type constitutive boundaries identified above would be more enriched for housekeeping genes than cell-type specific or cell-type common boundaries (present in 2–7 cell types). We measured the fold enrichment of

housekeeping genes at domain boundaries, as defined by the Human Protein Atlas[34]. This analysis reveals that cell-type-specific domain boundaries and boundaries present in 2 or 3 cell types are significantly depleted for the presence of housekeeping genes, and cell-type constitutive boundaries present in 7 cell types or constitutive to all 8 cell types, are significantly enriched for housekeeping genes (Fig. 7f).

## Discussion

Even though it is established that the architectural protein CTCF is a master regulator of chromatin conformation[35–38], the relationship between CTCF binding and regulation of long-range epigenome expression and chromatin conformation is not yet well established. Due to the ongoing debate about consequences of CTCF depletion on 3D architecture using different approaches and cell systems[13,14,39,40], we were particularly interested here to determine if CTCF binding also plays a role in the insulation of LRES and LREA regions[15,16]. We therefore explored the genome-wide chromatin effects of global CTCF depletion in prostate cancer cells to determine the role of CTCF in long-range epigenetically regulated domain organisation. We were surprised to find a conserved subset of CTCF sites that are resistant to CTCF depletion and propose that these persistent CTCF sites are essential for constitutive higher order chromatin architecture and the maintenance of long-range epigenetically regulated domains.

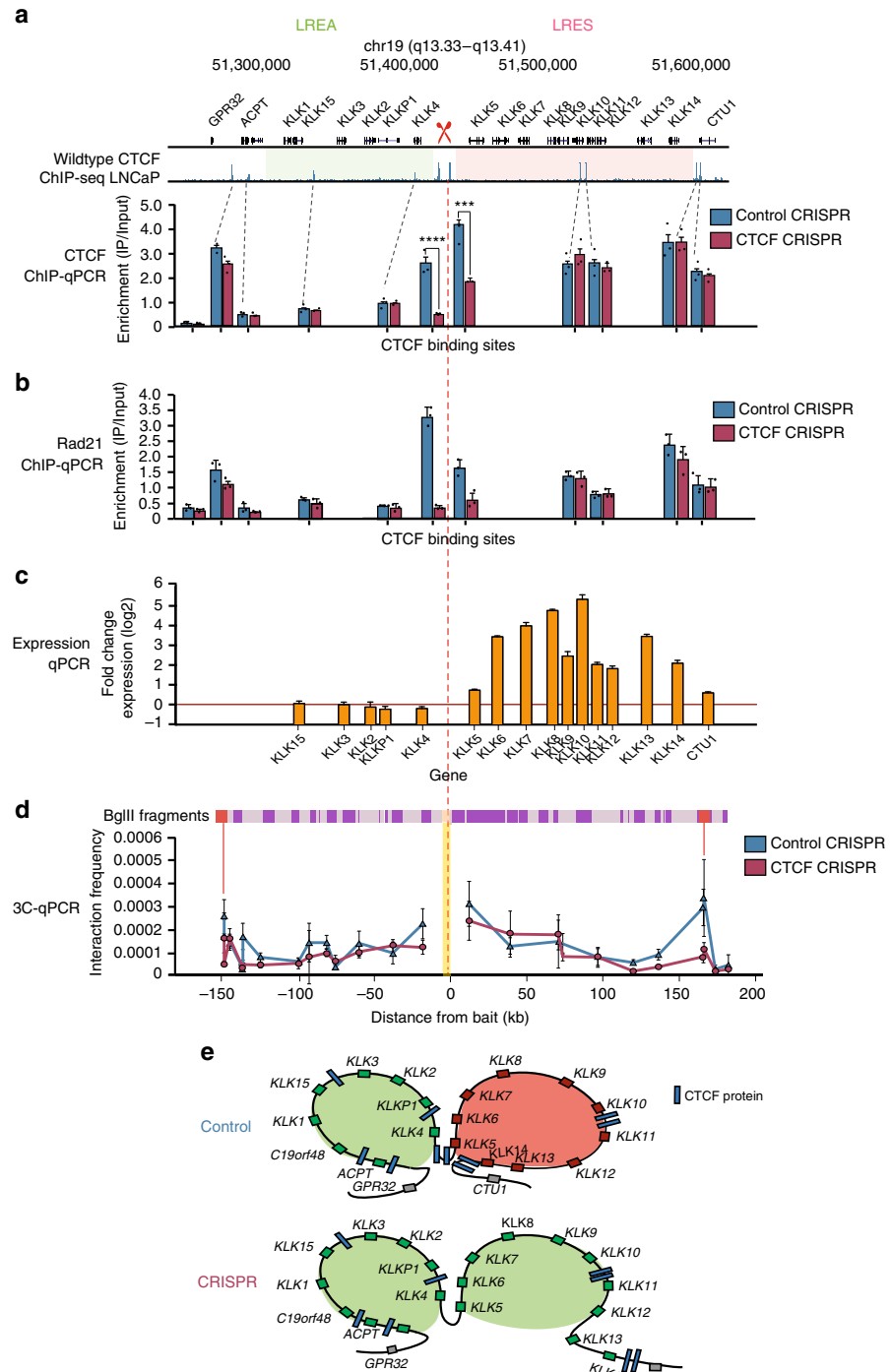

**Fig. 5 CRISPR of persistent CTCF sites results in coordinate up-regulation of LRES genes. a** CTCF ChIP-qPCR assaying all CTCF-binding sites at *KLK* locus was performed on control and CRISPR cells and demonstrates reduction of CTCF binding at CRISPR sites but retained binding at all other CTCF sites in the region (* indicates where two-tailed *t*-test, *p* < 0.001; *n* = 2 biologically independent experiments). Source data are provided as a Source Data file. Error bars represent SE. Data for technical replicates *n* = 3 are overlaid as a dot blot. **b** RAD21 ChIP-qPCR performed after CRISPR of persistent sites shows that cohesin binding is maintained at persistent sites but depleted across the other CTCF sites in the region. Error bars represent SE. Data for technical replicates *n* = 3 are overlaid as a dot blot. Source data are provided as a Source Data file. **c** qPCR for *KLK* gene expression following CRISPR of two persistent sites at the *KLK* locus shows coordinate up regulation of previously silenced genes. Error bars represent SE. Source data are provided as a Source Data file. **d** Biological replicate #1 of 3C-qPCR at the *KLK* locus following CRISPR of persistent sites shows changes to looping intensity between the bait fragment (yellow bar) and its upstream and downstream interacting fragments (red lines). Bars above the graph illustrate BglII fragments across the region. Error bars represent SE between technical replicates ((x3); see Supplementary Fig. 6a). **e** Schematic representation of data in **d** demonstrates the weakening of interaction between the bait and the downstream anchor point and expression of the normally silenced genes in the LRES region following CRISPR of CTCF-persistent sites.

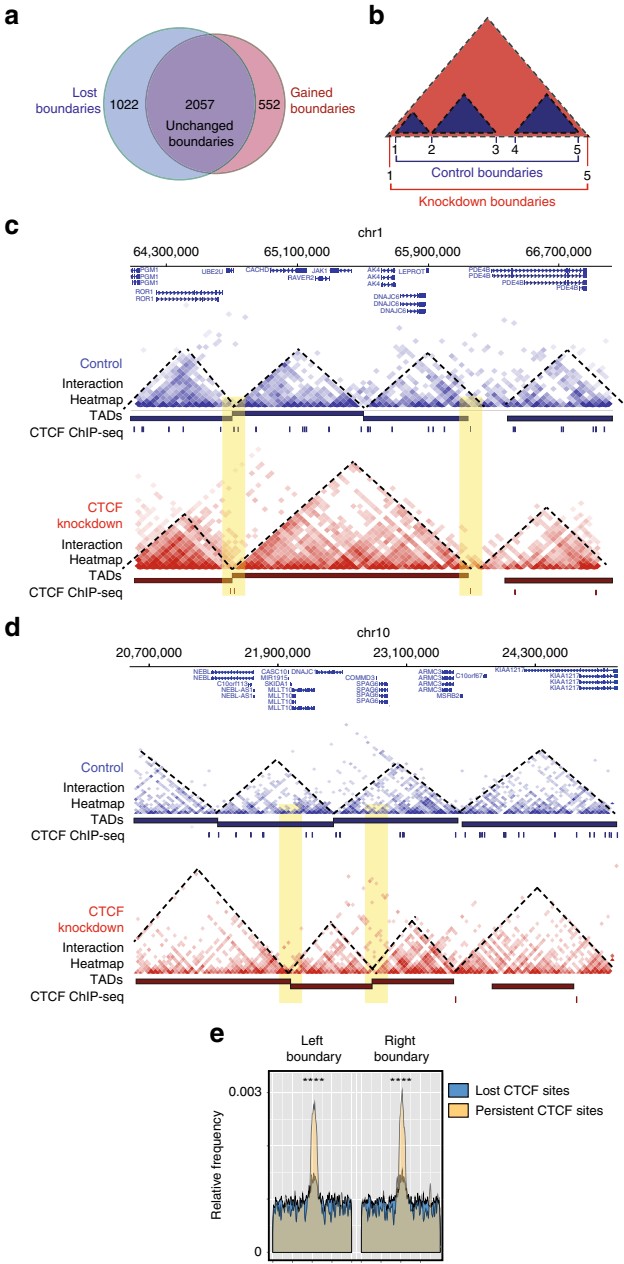

**Fig. 6 TAD boundaries are altered after CTCF RNAi in LNCaP cells. a**
Venn diagram demonstrates number of shared, lost and gained TAD
boundaries between control and CTCF knockdown conditions. **b** Schematic
illustrating TAD merging and loss/retention of TAD boundaries. **c**
Screenshot from Washu epigenome browser showing chromatin
interaction heatmaps, TADs and CTCF ChIP-seq data for 144 h control
(blue) and CTCF (red) RNAi conditions in LNCaP cells. At this locus, two
TADs become merged into one upon loss of CTCF binding across the
region. The larger, merged TAD has persistent CTCF binding at its
maintained boundaries. **d** Screenshot from Washu epigenome browser
showing chromatin interaction heatmaps, TADs and CTCF ChIP-seq data
for 144 h control (blue) and CTCF (red) RNAi conditions in LNCaP cells. At
this loci new TAD boundaries are formed upon loss of CTCF binding across
the region (highlighted in yellow). **e** Positional enrichment of all lost and all
persistent site binding at topological domain boundaries following CTCF
RNAi demonstrates that persistent CTCF sites are enriched at domain
boundaries that are present following knockdown (two-tailed $t$-test, $p <$
0.0001; $n = 2$ biologically independent experiments).

We performed ChIP-seq after genome-wide CTCF knockdown
in two different cell types and identified a small subclass of highly
conserved CTCF sites that remain persistently bound despite the
apparent robust loss of CTCF. Indeed we found that persistent
sites were commonly shared across different cell types unlike
lost sites that were more cell-type specific, in line with Schmidt
et al. [41,42]. Persistent CTCF sites, relative to all CTCF sites, are
typically less methylated and display more chromatin accessibility
and a higher intensity of CTCF binding suggesting that these
CTCF sites are more consistently bound across a population of
cells. Indeed, we found a remarkably high level of enrichment of
persistent CTCF sites at TAD boundaries that were conserved
across different cell types. This level of conservation suggests a
fundamental role for persistent CTCF sites in constitutive chro-
matin architecture. It has previously been reported that house-
keeping genes are generally enriched at TAD boundaries[2] but we
expanded this finding to show that enrichment of housekeeping
genes increases incrementally with the level of conservation of the
boundary. Together, these findings suggest that preferential
retention of persistent CTCF binding is important to preserve the
expression of housekeeping genes and cell viability.

We found that genome-wide depletion of CTCF resulted in the
disruption of TAD insulation in regions that were depleted in
persistent sites, in particular resulting in the merging of TADs. In
contrast, persistent sites were enriched at TAD boundaries that
remain intact following CTCF depletion. This supports our
finding that retention of CTCF binding plays a role in main-
taining TAD structure and thereby protecting against large-scale
changes to gene expression. Other groups studying the con-
sequences of CTCF depletion have previously reported opposing
results demonstrating either major disruption to TAD structure[14]
or only modest changes[13,39]. Potentially these differences are due
to the different levels of residual CTCF binding post knockdown.
Our data suggests that the vulnerability of TAD to CTCF inter-
ference is context dependent and that TAD boundaries that are
maintained, lost or gained after CTCF depletion depend on the
distribution of persistent CTCF sites at TAD boundaries.

Finally, we showed that genome-wide CTCF depletion had
little effect on gene expression levels or active chromatin dis-
tribution across the adjacent LRES and LREA domains in the
*KLK* gene locus in prostate cancer cells. Even though the majority
of CTCF sites across the locus were depleted, binding persisted at
two CTCF sites at the boundary of the LREA and LRES regions.
Strikingly, targeted CRISPR–Cas9n deletion of CTCF persistent
sites at the *KLK* locus resulted in a change in chromatin looping,
and coordinated activation of the *KLK* genes contained within the
LRES domain. These results demonstrate the persistent CTCF
sites can play a direct role in the insulation of long-range epi-
genetically regulated domains due to loss of chromatin looping
and potential spreading of gene activation from the LREA region.
Previous locus-specific studies have also shown CTCF protein can
play a role in mediating chromatin folding[7,43]. Multiple studies
have also shown that CTCF positions cohesin at CTCF sites
across the genome[26,44,45]. In accordance with this finding we
show that following CTCF knockdown at the *KLK* region, RAD21
also remains bound at persistent sites but is lost from all other
CTCF sites. This suggests that cohesin preserves its usual func-
tions at this location and thus it is likely that cohesin works in
concert with persistent CTCF to maintain boundaries.

Taken together the data demonstrates that a portion of CTCF
binding persists after CTCF depletion and that these sites are
conserved and located at constitutive TAD boundaries. This
supports a model where there is a preferential order to the loss of
TAD insulation upon CTCF knockdown, where chromatin

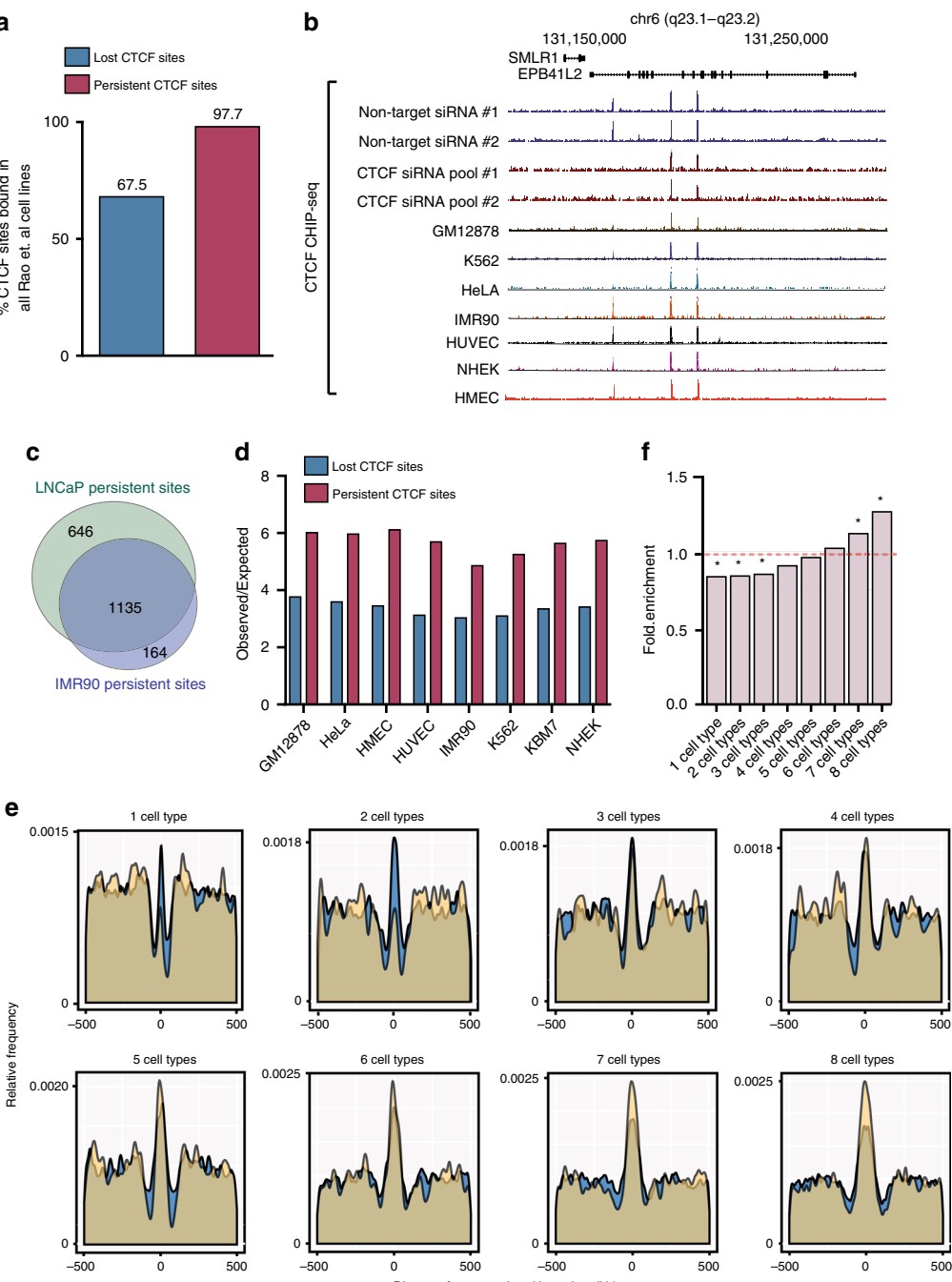

**Fig. 7 Persistent sites are constitutively bound by CTCF in other cell types. a** Bar graph shows greater proportion of persistent sites are constitutively bound by CTCF in GM12878, K562, HeLA, IMR90, HUVEC, NHEK and HMEC cell lines when compared to lost sites. **b** CTCF ChIP-seq data for replicate LNCaP RNAi experiments and wildtype GM12878, K562, HeLA, IMR90, HUVEC, NHEK and HMEC cells shows persistent sites are constitutively bound by CTCF in other cell types at an example locus. **c** Venn diagram illustrating overlaps of LNCaP-persistent sites with IMR90-persistent sites. **d** Graph showing observed/ expected ratio for lost (blue bars) and persistent sites (red bars) to overlap with domain boundaries in GM12878, K562, HeLA, IMR90, HUVEC, NHEK, HMEC and KBM7 cells. Comparing persistent to lost CTCF binding shows that persistent sites are more enriched to overlap domain boundaries. **e** Domain boundary data for GM12878, K562, HeLA, IMR90, HUVEC, NHEK, HMEC and KBM7 cells was subset based on whether each domain boundary was cell-type specific (present in 1 cell line), common (present in 2–7 cell lines) or cell-type constitutive (present in all 8 cell lines). Positional enrichment of lost and persistent CTCF sites was plotted for each subset. Lost sites are enriched at cell-type-specific domain boundaries. Persistent sites are enriched at cell-type constitutive domain boundaries (two-tailed t-test, *p = 0.02; n = 2 biologically independent experiments). **f** Enrichment of housekeeping genes at cell-type specific, cell-type common and cell-type constitutive TAD boundaries from Rao et al.[4] (Hyper-geometric test, *p-value < 0.05 after Bonferroni correction).

conformation fundamental to cellular viability and long-range gene expression is preferentially maintained. Our results highlight the importance of reinterpretation of previous CTCF depletion studies to determine the genomic landscape and location of the structural role of the residual CTCFs. Our results support a future systematic approach to dissect mechanisms regulating key CTCF-associated spatial control of chromatin architecture and long-range epigenetic-regulated domains.

## Methods

**Cell culture**. Prostate cancer cell line—LNCaP, and normal lung fibroblast cell line—IMR90, were obtained from the American Type Culture Collection (ATCC). All were cultured under recommended conditions at 37 °C and 5% $CO_2$.

**CTCF RNAi**. LNCaP were transfected with 20 nM of Dharmacon ON-TARGET plus pooled non-targeting siRNA (#D-001810-10-20, Dharmacon) or 20 nM of Dharmacon ON-TARGET plus pooled CTCF siRNA (#LU-020165-00-0010, Dharmacon) (containing sequences GAUGAAGACUGAAGUAAUG, GGAGAA AGAAGAAGAGUA, GAAGAUGCCUGCCACUUAC, GAACAGCCCAUAAAC AUAG) according to the manufacturer-supplied protocol. Following 72 h of transfection, cells were split and reverse transfected for a further 72 h (144 h). IMR90 cells were transfected with the aforementioned siRNAs and concentrations, however the transfection was performed using Lipofectamine RNAiMAX transfection reagent according to the manufacturer's instructions.

**Western blots**. After 72 and 144 h post-transfection nuclear protein was harvested by adding 10 mM Tris–HCl, 10 mM NaCl, 3 mM $MgCl_2$, 0.1 mM EDTA and 0.5% IGEPAL to the cell pellet followed by dounce homogenisation and centrifugation. The supernatant containing cytoplasmic protein was removed and the remaining pellet was incubated for 15 min on ice with 400 mN NaCl, 7.5 mM $MgCl_2$, and 0.2 mM EDTA. The suspension was centrifuged and supernatant containing nuclear protein was collected. Protein concentration was determined using Pierce BCA Protein Assay Kit. Membranes were developed using Amersham ECL Plus Western Blotting Detection Reagents and visualised using photographic film. Western blots used antibodies against CTCF (#07-729, Millipore) diluted 1:5000 and total H3 diluted 1:20,000 (#ab1791, Abcam).

**Affymetrix gene ChIP human gene 2.0 ST array**. Array carried out by the Ramaciotti Centre for Genomics. Analysis of Affymetrix expression arrays was performed using R version 3.2.5. Raw CEL files were read in, background corrected, quantile normalised and median-polished using RMA[46] as implemented in the "oligo" bioconductor package[47]. Quality control plots (intensity distributions, RLE and NUSE plots) were generated and visually inspected to ensure data was of consistent quality across the experiment. Probesets from the HuGene array were annotated with Ensembl gene and transcript identifiers using biomaRt[48]. Differential expression analysis was performed using limma[49]. The design matrix used in linear model fitting included the siRNAs used (CTCF or control), the time point after siRNA addition (24, 72 or 144 h) and the paired nature of the biological replicates in the time course. The contrast matrix was constructed to detect differences in gene expression between the CTCF and non-targeting siRNA samples at each time point using moderated $t$-statistics. Genes were deemed significantly up-regulated or down-regulated if the Benjamini–Hochberg-adjusted $p$-value was <0.05. Distance between differentially expressed genes transcription start sites and their closest LNCaP CTCF site was calculated using "distanceToNearest" function in the GenomicRanges Bioconducor package[50]. Each experiment was performed in biological duplicate.

**ChIP-qPCR and ChIP-seq**. Cells were scraped into ice-cold PBS containing protease inhibitors and collected by centrifugation at 500×$g$ for 5 min at 4 °C. Pellet was resuspended in ice-cold PBS and fixed with 1% formaldehyde for 15 min at room temperature. Fixation was quenched with addition of 125 mM (final concentration) glycine for 5 min at room temperature. Fixed cells were centrifuged at 500 × $g$ for 5 min at 4 °C and washed 2× with 10 mls ice-cold PBS containing protease inhibitors. Pellet was resuspended in 1.5 ml nuclei extraction buffer (10 mM Tris–HCl pH 7.5, 10 mM NaCl, 3 mM $MgCl_2$, 0.1 mM EDTA and 0.5% IGEPAL) per 10 × 10[6] cells and incubated on ice for 10 min. Suspension was dounced 10× with a tight dounce and centrifuged at 1000 × $g$ for 5 min at 4 °C. Pellet was washed 1× with ice-cold PBS-containing protease inhibitors and centrifuged. Pellet was resuspended in 1x sonication buffer (50 mM Tris–HCl pH 8, 1% SDS, 10 mM EDTA) and fragmented to ~300 bp using a Branson probe Sonifier. Fragmented DNA was resuspended in 1 ml IP dilution buffer (16.7 mM Tris–HCl pH 8, 0.01% SDS, 1% Triton X-100, 167 mM NaCl, 1.2 mM EDTA) per ChIP (each ChIP contains 5 × 10[6] cells) and pre-cleared with 30 μl of Salmon Sperm DNA/Protein A agarose–50% Slurry (#16-157, Millipore) per ChIP. 10 μg of the following antibodies were added per ChIP and incubated overnight—CTCF (#07-729, Millipore), RAD21 (#ab992, Abcam), H3K4me3 (#39159, Active Motif), H3K27ac (#39133, Active Motif). Antibody/protein complexes were bound to Salmon Sperm DNA/Protein A agarose–50% Slurry. The samples were pelleted and washed once with low salt buffer (2 mM EDTA, 0.1% SDS, 1% Triton X-100, 20 mM Tris–HCl pH 8.1, 150 mM NaCl) once with high salt buffer 2 mM EDTA, 0.1% SDS, 1% Triton X-100, 20 mM Tris–HCl pH 8.1, 500 mM NaCl), once with LiCl buffer (1 mM EDTA, 10 mM Tris–HCl pH 8.1, 250 mM LiCl, 1% sodium deoxycholate, 1% IGEPAL) and twice with TE buffer (1 mM EDTA, 10 mM Tris–HCl pH 8.1). Antibody/protein complexes were eluted off beads using elution buffer (1% SDS, 100 mM sodium bicarbonate). Crosslinks were reversed with NaCl. Protein and RNA were degraded with Proteinase K and RNAse A, respectively. Libraries for ChIP-seq were prepared using TruSeq ChIP sample Preparation Kit (#IP-202-900, Illumina). For ChIP-seq the resulting libraries were sequenced on

the Illumina HiSeq 2500 platform configured for 50-bp single-end reads. Bowtie1[51] was used to align ChIP-seq reads to hg19 allowing up to three mismatches, discarding reads mapping to multiple positions in the genome and removing duplicate reads. Peaks were called using MACS2[52] or Peak ranger software[53] and both bigwig (signal) and bed (peak calls) files were visualised using the UCSC genome browser[54]. For ChIP-qPCR presence of protein binding was assessed using primers listed in Supplementary Table 1. Each experiment was performed in biological duplicate.

**Low input ChIP-seq**. Active Motif Low-Cell ChIP-seq kit was used according to the manufacturers instructions to perform CTCF ChIP on 35,000 IMR90 cells in duplicate and to prepare libraries. Libraries were sequenced on Illumina Next-Seq500 platform configured for 75-bp single-end reads. Fastq file analysis was performed using a custom pipeline based on a modified version of guidelines for molecular identifier (MID) analysis for single-read sequencing (Active Motif). Fastq files were merged by a custom script, adapter trimmed using cutadapt and read quality was assessed using FastQC. Reads were aligned to the human (hg19) genome using BWA and the resulting SAM file was sorted and converted into BAM format. Duplicate reads were identified and removed using custom Perl script provided by Active Motif. The de-duplicated BAM was used to generate bigwig files and peak calls with MACS2. The peak output files from MACS2 were then converted to bigwig files for visualisation.

**CRISPR–Cas9n**. CRISPR–Cas9n (D10A double nickase mutant) editing was performed according to the Nature Method protocol by Ran et al.[27] with some modifications. sgRNAs were designed using the online CRISPR Design Tool (http://tools.genome-engineering.org). CTCF motifs within the ChIP-seq peaks were called using HOMER[55]. Input sequences (Supplementary Table 3) for the Design tool were obtained from the UCSC genome browser and included the HOMER CTCF motifs (highlighted in red) within the *KLK*-persistent sites. sgRNAs that were immediately adjacent to a protospacer adjacent motif (PAM) were designed for each persistent site. The BbsI-cloning site (5′ CACC 3′) was added to the 5′ end of the sgRNAs and oligos (Supplementary Table 4) were ordered and purchased from Integrated DNA Technologies. Paired 100 μM forward and reverse sgRNAs were phosphorylated and annealed with 1x T4 ligation buffer with ATP (#B0202S, NEB) and T4 PNK (# M0201S, NEB) using the following conditions: 37 °C for 30 min; 95 °C for 5 min; ramp down to 25 °C at 5 °C/min. Products were then diluted 1:200 in $ddH_2O$. Diluted sgRNA oligos were each cloned into 100 ng pSpCas9n(BB)-2A-GFP (# PX461, Addgene) using 1x Tango buffer (#BY5, Thermo Fisher), 10 mM DTT (#18080-093, Invitrogen), 10 mM ATP (#P0756S, NEB). 1 μl Fast Digest BbsI (#FD1014, Thermo Fisher) and 0.5 μl T7 ligase with 2x rapid ligation buffer (#M0318L, NEB). The reaction was incubated for a total of 60 min in a thermocycler on the following cycle: 37 °C for 5 min, 21 °C for 5 min. Plasmids were transformed into α-Select Chemically Competent Cells (#BIO-85027, Bioline). 2 μl of plasmids were added to 20 μl of ice-cold cells and incubated on ice for 1 h. The mixture was heat-shocked at 42 °C for 2 min in a water bath and returned immediately to ice for 5 min. 250 μl of lysogeny broth (LB) was added to the mixture and incubated at 37 °C for 1 h. Cells were then streaked onto LB agar plates containing ampicillin and incubated overnight at 37 °C. Three colonies were picked per sgRNA (to increase chances of having correct insert) and inoculated into a 5 ml culture of LB with 100 μg ml$^{-1}$ ampicillin. Cultures were grown overnight at 220 rpm at 37 °C. Plasmid DNA was isolated using QIAGEN Spin Miniprep kit (#27104, Qiagen) following the manufacturer's instructions. Plasmids were submitted to Australian Genome Research Facility for sequencing with U6-Fwd promoter (GACTATCATATGCTTACCGT) and inserts were verified before being used for transfection. 350,000 LNCaP cells were transfected using 1 μg plasmid and Lipofectamine 3000 (#L3000015, Life Technolgies). GFP-positive cells were sorted and pooled 24 h after transfection. Sorted cells were expanded and RNA and chromatin were harvested in biological duplicate. qPCR was used to determine changes in Kallikrein region gene expression. Primers are listed in Supplementary Table 5.

**3C and Hi-C**. Hi-C experiments were based on ref. [4] with some modifications. 10 × 10[6] single cells were collected and fixed with 2% methanol-free, formaldehyde for 5 min at room temperature. Reactions were quenched with glycine and incubated at room temperature for 5 min followed by an additional 10 min on ice. Cells were centrifuged for 3 min at 500 × $g$ then washed in ice-cold PBS and protease inhibitors followed by an additional centrifugation. Nuclei were extracted by incubation in 1 ml ice-cold nuclei buffer (10 mM Tris, pH 8, 10 mM NaCl, 0.2% IGEPAL, plus protease inhibitors) per 10 × 10[6] cells for 2 h on ice followed by dounce homogenisation (30× strokes). Nuclei were collected by centrifugation at 4 °C for 10 min at 2500×$g$ then washed twice in 1 × NEBuffer3.1 (#B703S, New England Biolabs). Nuclei were resuspended in ice-cold 1 × NEBuffer3.1 supplemented with 0.3% SDS then incubated at 37 °C for 1 h with gentle shaking. 1.6% Triton X-100 was added followed by as further 60 min incubation at 37 °C with shaking. Chromatin was digested overnight with 750U BglII (#R0144S, New England Biolabs) at 37 °C with shaking. Ends were repaired and marked with biotin-14-dATP using Klenow DNA polymerase (#M0210S, New England Biolabs) at 37 °C for 45 min. Samples were centrifuged for 5 min at 600×$g$ and all supernatant removed

except for 50 µl including the pellet. Ligations were performed in a final volume of 950 µl composed of T4 DNA Ligase (#M0292L, New England Biolabs), T4 DNA Ligase buffer, 1% Triton X-100 and 100 µg BSA. Reactions were performed at 18 °C for 4 h. Samples were centrifuged once more and supernatant removed before pellets were resuspended in 1000 µg Proteinase K, 1% SDS and 0.5 M NaCl and incubated at 65 °C overnight. The DNA was purified twice with phenol:chloroform: isoamyl alcohol 25:24:1 saturated with 10 mM Tris, pH 8.0, 1 mM EDTA. After the second extraction, DNA was precipitated with 80% ethanol, 50 mM sodium acetate and 10 µg glycogen overnight at −80 °C. DNA was collected by centrifugation at 18,000 × $g$ for 30 min at 4 °C and dissolved in nuclease-free water. Quality of digestion and ligation was assessed with gel electrophoresis. DNA was quantified using the Broad Range Qubit assay (#Q32854, Life Technologies). For 3C, ligated samples were assayed by qPCR in quadruplicate. In order to control for PCR efficiency, we digested and ligated BAC clones that spanned the entire length of the KLK region: CTC-771P3 (Start: Chr19, 51229459, End: Chr19, 51358173) and CTD-2342A18 (Start: Chr19, 51366253, End: Chr19, 51635598) and confirmed equal efficiency for primers with positive interactions. 3C primers are listed in Supplementary Table 2. For Hi-C, samples were taken forward to library preparation.

**Preparation of Hi-C libraries**. Hi-C libraries were prepared using a customised protocol described by Taberlay et al.[56]. Hi-C material was sonicated using a Covaris Focused-Ultrasonicator M220 instrument to achieve fragment sizes of 300–500 bp. Fragmented DNA had ends repaired using the NEBNext DNA Prep Master Mix Set for Illumina (#E6040L, NEB). Following end repair a size selection was performed using 1.6× volume AMPureXP Beads (#A63881, Beckman Coulter Inc.). Next blunt ends were dA-tailed using the NEB# E6040L kit. This material was added 1:1 to MyOne Streptavidin C1 beads (#650-01, Invitrogen) that had been resuspended in 2× binding buffer (10 mM Tris–HCl pH 8.0, 1 mM EDTA and 2 M NaCl) and incubated for 20 min at room temperature, with rotation. Biotin-tagged DNA coupled with MyOne Streptavidin C1 beads was isolated using a magnetic particle concentrator. Beads were washed once with 200 µl 1× binding buffer and resuspended in a final volume of 60 µl of $H_2O$. Adapters were ligated using the NEBNext Ultra DNA Library Prep kit (#E7370L, NEB). Adapter-ligated DNA was washed 2× with 200 µl of 1× tween wash buffer (5 mM Tris–HCl pH 8.0, 0.5 mM EDTA, 1 M NaCl and 0.05% Tween (#p7949, Sigma Aldrich)), 1× with 200 µl binding buffer and 1× with 200 µl 1× NEB2 buffer (#B7002S, NEB) before beads were resuspended in 30 µl $H_2O$. PCR enrichment of adapter-ligated DNA was performed on DNA bound to the MyOne Streptavidin C1 beads using NEB kit (# E7370L, NEB). The PCR cycling steps were 1 cycle at 98 °C for 30 s, 10–14 cycles at 98 °C for 10 s/65 °C for 75 s, 1 cycle at 65 °C for 5 min. Clean up of PCR products was performed using 1x volume AMPureXP Beads (#A63881, Beckman Coulter) and products were eluted into 30 µl of nuclease-free $H_2O$. Libraries were quantified using the KAPA Library Quantification Kit for Illumina platforms (#KP-KK4835, Geneworks). Library-size quality was assessed using a Bioanalyzer 2100 (Agilent Technologies). Libraries were sequenced (100 bp paired-end reads) on the HiSeq 2500 (Illumina) according to the manufacturer's instructions. We generated $30 \times 10^6$ and $101 \times 10^6$ valid, mapped read-pairs for the control and CTCF knockdown samples, respectively. Fit-Hi-C was used to pool biological replicates for downstream analysis.

**Normalisation of Hi-C data**. Performed as in Taberlay et al. [56]. All Hi-C-seq libraries were processed through the NGSane framework v0.5.2[57] available from Github using the "fastqc", "hicup" and "fithicaggregate" modules as follows: First, quality check of sequence libraries was performed with FastQC v0.11.2. Raw fastq files were then pre-processed, mapped with bowtie v1.1.0[51] and assessed for artefact levels through HiCuP v0.5.2 supplying genome assembly (hg19) and the BglII-restriction enzyme cut site. Aligned read files in BAM format were sorted with Samtools v1.2[58] and duplicates were tagged using MarkDuplicates from Picard tools v1.121. Replicates were pooled using bespoke Python scripts (provided within NGSane) leveraging the sparse matrices formats in the SciPy libraries (http://www.scipy.org/). Significant connections were assessed from contact count matrices at a 40 kb resolution using a custom adaptation of FitHi-C[59,60] (provided within NGSane) supplying iteratively corrected bias offsets calculated through HiCorrector v1.1[61] as well as genome mapability tracks from ENCODE. Significant contacts with false discovery rate (FDR) <0.01 were imported into the WashU Epigenome Browser[62] for visualisation and further analysis.

**Identification of topologically associating domains**. Performed as in ref. [56]. TADs were identified in the Hi-C-seq data using the 'domaincaller' pipeline developed by Dixon et al. [2]. The domaincaller algorithm uses a statistic called the 'directionality index' (DI) that quantifies upstream and downstream interaction bias of 40 kb genomic bins. This was developed based on the observation that the most upstream portion of a TAD is highly biased towards interacting downstream, and the downstream portion of a topological domain is highly biased towards interacting upstream. Hence the DI can be used to locate TADs and TAD boundaries. A Hidden Markov model (HMM) is used to determine the underlying directionality bias and uses the assumption that the DI is a mixture of "Upstream Interaction bias", "Downstream Interaction bias" or "No bias". These HMM calls are used to infer a TAD as a region beginning at a single downstream-biased HMM state, which continues through any contiguous downstream-biased states and ends

at the last in a series of upstream states. "Domain boundaries" were called as regions within 100 kb from the TAD and "unorganised chromatin" were regions at least 100 kb from the TAD.

**Visualisation of Hi-C data**. Performed as in Taberlay et al. 2016[56]. To visualise the segmentation of the interaction data into domains, we generated 2D heat maps at 40 kb resolution and overlaid them with previously generated ChIP-seq tracks and TAD tracks generated as BED files. Interaction frequencies were calculated as previously described and visualised in the WashU Epigenome Browser[62]. Positive score thresholds were adjusted manually to normalise for sequencing depth difference between control and CTCF RNAi samples.

**Domain boundary-enrichment analysis**. To determine if domain boundaries were associated with a given factor (histone marks, CTCF and RAD21), we used ngsplot (v2.47.1)[63] and plotted the averaged data around the ±500 kb region of the boundary. Additionally, we defined the percentage of overlap between CTCF and H3K4me3 binding sites and domain boundaries by intersecting peaks identified from ChIP-seq data with the domain boundaries.

**Differential histone mark binding**. DiffBind[64] was used to identify differential binding of H3K4me3 and H3K27ac. Peaks were called in each treatment group and replicate separately using PeakRanger[53], merged into a consensus peakset for the experiment retaining only peaks that occurred in two or more samples, the number of aligned sequencing reads occurring in each consensus peak in each sample was then counted. Potential differential binding between the treatment groups was then assessed using the edgeR method[65] and significantly differentially bound peaks called using an FDR cutoff of 0.01, minimum absolute log2 fold-change of 1 and minimum peak height of 7 in at least one condition.

**Observed/expected enrichment calculation**. The observed/expected ratio to assess the enrichment of the overlap of two sets of genomic regions compared to genomic background was calculated by assessing the degree of overlap between the "test" and "query" regions using the "findOverlaps" function in the GenomicRanges Bioconductor package[50], and then randomly shuffling the "query" regions throughout the genome (restricted to autosomes and sex chromosomes) 1000 times using "shuffleBed" from the BEDTools package[66] with the options "-maxTries 10,000 -noOverlapping".

**Housekeeping gene enrichment at arrowhead boundaries**. A hyper-geometric test was used to test for enrichment of housekeeping genes (Human Protein Atlas[34]) within sets of genes found within 100 kb of Arrowhead domain borders[4] separated by the numbers of cell types they were present in (one– eight cell types). More specifically, the overlap between housekeeping genes and gene sets was computed and compared to what would be expected by chance if equivalent number of genes were drawn uniformly at random from the background set of genes. We report statistically significant enrichments with Bonferroni-corrected p-value < 0.05.

**Statistical tests for positional-enrichment graphs**. Welch two-sample t-test was performed on the calculated average 'distance to feature' for persistent and lost subsets.

**Reporting summary**. Further information on research design is available in the Nature Research Reporting Summary linked to this article.

## Data availability
The data that support this study are available from the corresponding author upon request. The datasets generated and/or analysed during the current study have been uploaded to the Gene Expression Omnibus repository, GEO number GSE125641.

The source data underlying Figs. 3c, 4c and 5a–c and Supplementary Figs. 5A and 11A are provided as a Source Data file.

## Code availability
The custom code used to analyse the data is available from the corresponding author upon request.

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

## Acknowledgements

We thank members of the Clark Laboratory for helpful discussions and reading of the manuscript. We thank the Ramaciotti Centre for Genomics for performing the Affymetrix array. This work was supported by project (APP1147974; APP1070418) and fellowship (APP1156408) grants from the National Health and Medical Research Council (NHMRC) to S.J.C. The contents of the published material are solely the responsibility of the administering institution and individual authors and do not reflect the views of the NHMRC.

## Author contributions

Conception and design: S.J.C. conceived and supervised the project. Experimental data: A.K. performed all the experiments. S.A.B. designed the 3 C assay. G.C.S. assisted in low input ChIP-seq experiments. F.V.-M. provided advice about performing low input ChIP-seq. P.T. provided initial guidance in performing 3 C assay. A.J.P. performed initial CTCF knockdown optimisation experiments. C.S. provided experimental advice. Data Analysis: A.L.S. carried out the bioinformatics analysis. Hi-C processing steps using fithic and domaincaller were performed by JAK. P.-L.L. processed and mapped low input ChIP-seq data. H.J.F. performed Affymetrix array analysis. Q.D. performed housekeeping gene-enrichment analysis. T.J.P. performed some statistical analysis. Writing, reviewing of manuscript: A.K., S.J.C.

## Competing interests

The authors declare no competing interests.
