## [Peer Review File · Nature Communications]

Reviewers' comments:

Reviewer #1 (Remarks to the Author):

In this manuscript the authors present data demonstrating that RNAi knockdown of CTCF does not compromise organization of the genome into TADs. The authors explain it by the observation that CTCF sites located at the TAD borders bind CTCF much stronger than the other ones. They also showed that a partial deletion of CTCF binding motives at the border of two TADs within KLK locus resulted in a loss of functional insulation of the TAD harboring repressed genes.

It should be noted that most of the results presented in the MS are confirmatory rather than novel. Indeed, it has been shown previously that a depletion of CTCF by RNAi is not sufficient to compromise TAD architecture (Ref 13 in this MS) while full degradation of CTCF results in a loss of TADs (Ref 14 in this MS). The fusion of TADs caused by targeted deletion or inversion of CTCF sites at TADs border also was reported. For this reason I do not think that this manuscript fits the criteria of Nature Communication. After addressing the issues specified below the authors may consider a possibility to submit it to a more specialized journal.

Technical comments:

1. Fig 1B. In the figure the regions interacting with the anchor placed between the two TADs are located at ~ -145 and $\sim +165$ from the anchor. However, in the text the authors state: We found two interactions; one occurred 129.5kb upstream of the bait and the other 183.8kb downstream from the bait. This should be corrected
2. The interactions identified in 3C analysis (Fig 1B) should normally be verified by reciprocal experiments with the anchors placed on the identified peaks. The authors should perform such experiment. Besides confirmation of the observed interactions this experiment will also show if the upstream and the downstream borders of two loops are located in a proximity, i.e. if there is an interaction between ~ -145 and $\sim +165$ regions.
3. The authors should indicate the direction of identified CTCF sites, as it was reported previously that, in mammalian cells, the loop bases mostly contain convergent CTCF sites.
4. Figure 3A is misleading. What does it show? One representative peak for each case? If so, is it not better to show bar-plots illustrating the figures indicated below the peaks?
5. Figure 3B. It appears that in cells treated with CTCF -specific RNAi the CTCF is no longer deposited at the upstream and the downstream borders of KLK cluster. What happens with the loops? It is reasonable to repeat in these conditions the 3C experiments presented in Figure 1B. In fact, it is also interesting to study TADs structure using a capture-C protocol.
6. Figure 4. The level of CTCF binding sites removal is rather moderate (~ 80 and $\sim 50\%$). Yet, the biological consequences are quite obvious. The authors should try to present some explanation. Interestingly, the upstream (repressed) loop is almost non-affected (12,6% reduction of interaction) while the downstream loop is almost destroyed (66% reduction of interaction). I wonder while in Fig 4E one see that both the upstream and the downstream region of then cluster are organized into loops. Perhaps, it is better to present the downstream area as having a linear configuration. Also, if the upstream area is still organized in a loop and thus is supposedly insulated, how the activation signal is spreaded from this loop into an inactive part of the cluster? To answer this question it will be helpful to know the actual levels of gene expression (not just the fold change). Do these levels become comparable after deletion of CTCF sites or they are still much higher in the upstream part of the cluster?
7. In Fig. 4E the distance between the upstream and the downstream ends of the KLK cluster become larger after deletion of internal CTCF sites. It is necessary to check this using 3C with anchors of the upstream and the downstream ends of the cluster. Perhaps, after deletion of internal CTCF sites, the two loops become fused into one large loop. The above-mentioned 3C experiment will clarify this possibility.

Reviewer #2 (Remarks to the Author):

In this study Clark and colleagues have attempted to address an outstanding question in the field on the impact of CTCF depletion on topologically associated domain (TAD) maintenance and gene regulation. To do this, they have generated siRNA knockdown of CTCF in the LNCaP cell line and examined how the knockdown affects genome-wide CTCF binding using ChIP-seq and genome-wide chromatin conformation using HiC. They have also use the KLK locus as an in-depth example to illustrate the impact of their CTCF knockdown experiments. The key finding of the study is that a subset of CTCF/cohesin binding sites are particularly persistent and these play an important role in the maintaining TADs that are conserved across cell types. Although the study does not really provide much new knowledge about CTCF chromatin structure regulation in a physiological context, the results are nevertheless important because the identification of these persistent CTCF binding sites possibly explains recent studies that report conflicting consequences of CTCF depletion and furthers our understanding of the role of CTCF in TAD maintenance. Overall, the results are clearly presented and are generally supported by the data. I have some specific comments below:

(1) My main concern with the study is the impact of knockdown efficiency on the number of persistent CTCF sites identified. The much larger number of persistent CTCF sites in IMR90 illustrate this. While I don't think the set of persistent sites in LNCaP are incorrect, the issue with the dependence on knockdown efficiency is that the overall results of the study may change quite significantly with a change in knockdown efficiency. For instance, if all the analysis of the study was done using the results from IMR90, the results may be somewhat different. As such, I think the authors should be more cautious in concluding that there is a subset of essential (presumably persistent) CTCF binding sites important in constitutive higher order chromatin maintenance. Many lost sites are also highly conserved and probably also essential. The "essentialness" as defined by their persistent sites is likely depend on their knockdown efficiency and I don't think the results are representative of true set of biologically "essential" CTCF binding sites. As such the persistent CTCF sites as presented throughout the paper can only be considered an enrichment. If the authors want to identify a true set of essential sites, they would likely need to show that the persistent set of sites become more consistent over a particular knockdown efficiency level.

(2) Can the authors explain the role of the lost CTCF/cohesin sites within LREA and LRES (For example the sites near KLK9 and KLK10)? Do they form internal "TADs" within the LRES or interact with other CTCF binding sites outside the LRES?

(3) Why is there still CTCF binding after CRISPR-Cas9 deletion of the motifs? Is it due to incomplete deletion? Or just persistent CTCF binding without the motif?

(4) The error bars seem to be very large for LRES TAD boundary in Fig 4C. Since it is only duplicates for each data point, can the independent replicates values be shown instead of the error bar? Also why is the interaction frequency scale so different to Fig 1C?

(5) If the goal is to show that the persistent sites have distinct chromatin characteristics, it would be a fairer comparison to only compare persistent and lost CTCF peaks with similar CTCF levels the NOME-seq and motif enrichment analysis. The lost peaks with weaker CTCF binding can be expected to lose CTCF binding first upon CTCF depletion in any case.

Reviewer #3 (Remarks to the Author):

In this manuscript, Khoury and colleagues describe the impact of CTCF decrease on gene expression and chromatin architecture. In line with work from others (Schmidt et al., 2012 Cell), knockdown of CTCF resulted in a selective loss of specific CTCF sites, while other regions were not affected. The 'persistent' CTCF sites were generally shared between different cell types, while the

CTCF sites lost after siRNA were more cell type specific, again in line with Schmidt et al. Much to my surprise, this other manuscript is not cited or mentioned. As novel findings, it is now shown that depletion of CTCF resulted in alteration in the TAD structure. The 'persistent' CTCF sites were at TAD boundaries that were not altered after knockdown. Altogether, I find this an interesting manuscript, with novel interesting findings regarding the impact of CTCF on the 3D genome architecture. That said, novelty in relation to the 2012 Schmidt et al Cell paper is a matter of debate, as many of the claims (for example the entire Figure 7) have already been described by others.

Specific comments:

Page 7, line 158:

The authors state there are a subset of 'persistent' CTCF sites to siRNA CTCF. In 2012, Schmidt et al. Cell identified highly conserved CTCF sites across species, which were resistant to knockdown. As such, many of the conclusions drawn in the current work have also been reported by Schmidt et al. (practically the entire Figure 7), yet that paper is not cited in the current manuscript. The authors should include information of the overlap of their 'persistent' CTCF sites and these conserved sites (human), and place their findings into context with the data reported by Schmidt et al.

Page 7, line 162: There is discussion of 'persistent' sites and 'lost' sites, but not of gained sites. Are there gained sites? And if so, what are the characteristics of these sites? Reprogramming of CTCF in response to knockdown may be important biologically.

Page 9, line 218: A measurement of sequence conservation like PhastCons might also be revealing here between persistent and lost sites.

Page 14, line 335: The claim is too strong that the alteration of TAD boundary structure occurs only at lost CTCF sites. Is there any statistical analysis to back this statement up?

Figure 4D: It's not clear to me how the authors conclude a 66.1% reduction in chromatin interaction strength. The lines are mostly overlapping, with the exception of the site in the final gene, CTU1. What's the basis of the 66,1% reduction conclusion?

Figure 5A. Include standard error shading in the profiles.

Figure 5C. Zoom in is necessary to see differences. As far as I can tell, the methods for NOME-seq are not included. As this method relies on bisulfite conversion, a control sample needs to be shown as well.

We appreciate the reviewer's time and constructive comments. We have performed many additional 3C experiments, as well as repeated the siRNA CTCF in IMR90 cells to ensure similar levels of CTCF reduction. In addition we provided new analyses of the persistent CTCF sites and revised the Manuscript accordingly. We have modified our Figures, expanded the Methods section and provided additional Supplementary Figures. We believe that the new data and analyses have further strengthened our manuscript and have addressed the reviewer's concerns.

We have marked changes in the manuscript in blue.

Please find a summary of our responses below:

Reviewer #1 (Remarks to the Author):

In this manuscript the authors present data demonstrating that RNAi knockdown of CTCF does not compromise organization of the genome into TADs. The authors explain it by the observation that CTCF sites located at the TAD borders bind CTCF much stronger than the other ones. They also showed that a partial deletion of CTCF binding motives at the border of two TADs within KLK locus resulted in a loss of functional insulation of the TAD harboring repressed genes.

It should be noted that most of the results presented in the MS are confirmatory rather than novel. Indeed, it has been shown previously that a depletion of CTCF by RNAi is not sufficient to compromise TAD architecture (Ref 13 in this MS) while full degradation of CTCF results in a loss of TADs (Ref 14 in this MS). The fusion of TADs caused by targeted deletion or inversion of CTCF sites at TADs border also was reported. For this reason I do not think that this manuscript fits the criteria of Nature Communication. After addressing the issues specified below the authors may consider a possibility to submit it to a more specialized journal.

Author Response:

Previous Knowledge in the Field

Several studies have yielded *inconsistent* results on the role of CTCF in long-range chromatin interactions and in the maintenance of TAD boundaries. For example CRISPR-Cas9 depletion of CTCF located at candidate TAD boundaries has been reported to be sufficient to deplete the targeted boundary (Narendra et al 2015, Lupianez et al. 2015). However a 2018 study by Barutcu et al. showed that deletion of a CTCF-rich locus at a TAD boundary did not cause loss of that boundary. Moreover a CTCF siRNA approach reported a general maintenance of TAD boundaries and modest changes to gene expression (Zuin et al., 2014), however no ChIP-seq was performed to assess the actual disruption and loss of CTCF sites. In comparison, Nora et al., 2017 used the AID approach, but ChIP-seq showed that 27% CTCF binding was retained however they still observed complete loss of insulation at 80% of TAD boundaries. In our study we show that even with more complete loss of CTCF sites (ie only 11.6% of sites retaining CTCF binding), we only observe loss of ~20% of TAD boundaries. In Wutz et al. 2017 the AID system was again utilised and showed that CTCF degradation did not change contact frequencies within or across TADs, but that TAD borders became 'fuzzier'. Importantly, another research group has also performed CTCF depletion using the AID system in mESCs (same cells used

in Nora et al. 2017) and found the opposite to Nora et al. i.e. almost complete maintenance of TADs (preprint - <https://www.biorxiv.org/content/early/2017/03/20/118737>).

Taken together, it is clear that a further investigation of the effects of CTCF depletion on 3D architecture is needed.

In addition the role of CTCF in the establishment or maintenance of long-range epigenetic silencing (LRES) and activation (LREA) has not been previously addressed.

New Insights and Conceptual Advances

Here, we explore the genome-wide chromatin effects of global (80-90%) CTCF depletion and CRISPR-targeted CTCF deletion and show that there is a core set of CTCF sites that are resistant to CTCF depletion. Importantly we show that these persistent sites are highly conserved between different cell types with 87% overlap between LNCaP and IMR90 cells following CTCF siRNA. Characterisation of these sites show that they are enriched at TAD boundaries, resistant to perturbation and are associated with stronger CTCF binding.

We propose these persistent CTCF sites are essential for cell-type constitutive higher order chromatin architecture and the maintenance of long-range epigenetically regulated domains. Our data suggests that the vulnerability of TAD to CTCF interference is context dependent and that TAD boundaries that are maintained, lost or gained after CTCF depletion depend on the distribution of persistent CTCF sites at TAD boundaries.

Importantly, we show that persistent CTCF sites play a mechanistic role in demarcation of the LREA and LRES domains across the Kallikrein locus in prostate cancer cells.

Together our data demonstrates that there is a subset of essential CTCF binding sites, which are involved in cell-type constitutive higher order chromatin architecture and modulation of these exemplary sites results in localised changes to gene expression across large epigenetically regulated domains.

Technical comments:

1. Fig 1B. In the figure the regions interacting with the anchor placed between the two TADs are located at ~ -145 and $\sim +165$ from the anchor. However, in the text the authors state: We found two interactions; one occurred 129.5kb upstream of the bait and the other 183.8kb downstream from the bait. This should be corrected

Response:

The distances between the bait and the interacting fragments have been corrected in the text on page 5.

2. The interactions identified in 3C analysis (Fig 1B) should normally be verified by reciprocate experiments with the anchors placed on the identified peaks. The authors should perform such experiment. Besides confirmation of the observed interactions this experiment will also show if the upstream and the downstream borders of two

loops are located in a proximity, i.e. if there is an interaction between ~-145 and $\sim+165$ regions.

Response:

Thankyou for the suggestion. We have now performed the reciprocate experiments (x3) using the upstream and downstream interacting fragments as baits (**new Figure 2 and Supp. Fig 1**). The results support a looping structure whereby LREA and LRES regions form independent loops from each other. The reciprocal 3C data also shows a weaker interaction between the fragments located at ~-149.2 kb and $\sim+163.8$ kb from the original bait which suggests that there is relatively little interaction between the outer borders of the KLK domains. We have modified our Figures and text accordingly on page 5-6.

3. The authors should indicate the direction of identified CTCF sites, as it was reported previously that, in mammalian cells, the loop bases mostly contain convergent CTCF sites.

Response:

The directions of the CTCF motifs (determined using HOMER) have now been added to Figure 2. CTCF binding motifs flanking chromatin loops mostly have a convergent orientation (Rao, 2014; Rao, 2017). The downstream loop at the KLK locus has anchors with convergent CTCF motifs, however the upstream loop is potentially anchored by divergent CTCF motifs (**new Figure 2A**).

4. Figure 3A is misleading. What does it show? One representative peak for each case? If so, is it not better to show bar-plots illustrating the figures indicated below the peaks?

Response:

We apologise if this Figure is misleading. We have now changed it to a bar plot as suggested by the reviewer (**new Figure 4A**).

5. Figure 3B. It appears that in cells treated with CTCF –specific RNAi the CTCF is no longer deposited at the upstream and the downstream borders of KLK cluster. What happens with the loops? It is reasonable to repeat in these conditions the 3C experiments presented in Figure 1B. In fact, it is also interesting to study TADs structure using a capture-C protocol.

Response:

We have now performed 3C experiments in duplicate on the CTCF siRNA cells and vector control. The results (shown in **new Supplementary Figure 2**) indicate that the upstream and downstream KLK cluster interactions are reduced (average 16.9% and 36.3% respectively) but not totally obliterated, suggesting that these CTCF sites are not as critical for the loop formation or gene expression control, in comparison to the internal CTCF sites. See new results page 7.

We agree that it would be interesting to study the TAD structure using a capture C protocol but with all the additional experiments we have already performed we believe that this is outside the scope of this study.

6. Figure 4. The level of CTCF binding sites removal is rather moderate (~80 and ~50%). Yet, the biological consequences are quite obvious. The authors should try to present some explanation. Interestingly, the upstream (repressed) loop is almost non-affected (12,6% reduction of interaction) while the downstream loop is almost destroyed (66% reduction of interaction). I wonder while in Fig 4E one see that both the upstream and the downstream region of then cluster are organized into loops. Perhaps, it is better to present the downstream area as having a linear configuration. Also, if the upstream area is still organized in a loop and thus is supposedly insulated, how the activation signal is spreaded from this loop into an inactive part of the cluster? To answer this question it will be helpful to know the actual levels of gene expression (not just the fold change). Do these levels become comparable after deletion of CTCF sites or they are still much higher in the upstream part of the cluster?

Response:

As suggested we have now included the actual levels of gene expression changes in **new Supplementary Figure 5**. While the LRES genes are co-ordinately increased in expression, they are consistently lower in expression than the LREA genes. Indeed the upstream loop remains more intact following the CRISPR of the boundary CTCF sites and thus the increased expression of the LRES genes appears to be associated more with loss of its own looping structure and chromatin interactions at the downstream border. However, activation of the LRES genes could be additionally influenced by partial spreading of gene activation from the LREA region due to the partial reduction of its chromatin loop structure. Please see revised discussion on page 16.

We have modified schematic representation to show a loosening of the loop structure but did not represent the LRES as a linear conformation as our data suggests that this also is not completely obliterated (**new Figure 5E**).

7. In Fig. 4E the distance between the upstream and the downstream ends of the KLK cluster become larger after deletion of internal CTCF sites. It is necessary to check this using 3C with anchors of the upstream and the downstream ends of the cluster. Perhaps, after deletion of internal CTCF sites, the two loops become fused into one large loop. The above-mentioned 3C experiment will clarify this possibility.

Response:

We have now used the upstream 3C bait as anchor on the CRISPR material (see **new Supplementary Figure 6B**). Our new data does not show a change in interaction strength between the outer borders of the KLK locus, which suggests that the two loops do not become fused into one large loop (page 9).

Reviewer #2 (Remarks to the Author):

In this study Clark and colleagues have attempted to address an outstanding question in the field on the impact of CTCF depletion on topologically associated domain (TAD) maintenance and gene regulation. To do this, they have generated siRNA knockdown of CTCF in the LNCaP cell line and examined how the knockdown affects genome-wide CTCF binding using CHIP-seq and genome-wide chromatin conformation using HiC. They have also use the KLK locus as an in-depth example to

illustrate the impact of their CTCF knockdown experiments. The key finding of the study is that a subset of CTCF/cohesin binding sites are particularly persistent and these play an important role in the maintaining TADs that are conserved across cell types. Although the study does not really provide much new knowledge about CTCF chromatin structure regulation in a physiological context, the results are nevertheless important because the identification of these persistent CTCF binding sites possibly explains recent studies that report conflicting consequences of CTCF depletion and furthers our understanding of the role of CTCF in TAD maintenance. Overall, the results are clearly presented and are generally supported by the data. I have some specific comments below:

(1) My main concern with the study is the impact of knockdown efficiency on the number of persistent CTCF sites identified. The much larger number of persistent CTCF sites in IMR90 illustrate this. While I don't think the set of persistent sites in LNCaP are incorrect, the issue with the dependence on knockdown efficiency is that the overall results of the study may change quite significantly with a change in knockdown efficiency. For instance, if all the analysis of the study was done using the results from IMR90, the results may be somewhat different. As such, I think the authors should be more cautious in concluding that there is a subset of essential (presumably persistent) CTCF binding sites important in constitutive higher order chromatin maintenance. Many lost sites are also highly conserved and probably also essential. The "essentialness" as defined by their persistent sites is likely depend on their knockdown efficiency and I don't think the results are representative of true set of biologically "essential" CTCF binding sites. As such the persistent CTCF sites as presented throughout the paper can only be considered an enrichment. If the authors want to identify a true set of essential sites, they would likely need to show that the persistent set of sites become more consistent over a particular knockdown efficiency level.

Response:

We appreciate this comment and agree that the impact of knockdown efficiency may affect the number of shared persistent sites identified across the 2 cell lines. We therefore have now performed optimisation experiments to increase CTCF knockdown efficiency in IMR90 cells from ~60% mRNA depletion to ~80% CTCF mRNA depletion (**new Supplementary Figure 11A**). We found that increasing the efficiency of CTCF knockdown in the IMR90 cells also caused a decrease in cell viability and this may explain why we could not get to an equivalent 90% depletion. The revised IMR90 conditions have been updated in the Materials and Methods section.

The ~80% CTCF knockdown in IMR90 cells reduced the number of persistent CTCF sites in the knockdown cells to 1840. Remarkably 87.3% of these retained CTCF sites in IMR90 cells overlapped the subset of persistent sites identified in LNCaP cells (**new Figure 7C**). The new data further supports our conclusion that persistent sites become more consistent over similar knockdown efficiencies and that these sites are more likely to be important for cell viability. Please see page 13 of the manuscript.

(2) Can the authors explain the role of the lost CTCF/cohesin sites within LREA and LRES (For example the sites near KLK9 and KLK10)? Do they form internal

“TADs” within the LRES or interact with other CTCF binding sites outside the LRES?

Response:

As suggested by Reviewer 1 we have now performed 3C experiments using the upstream and downstream interacting fragments also as baits. This revealed that the sites near KLK10 make contact with the downstream bait and therefore form an internal loop (**new Figure 2C and Supplementary Fig 1C**), page 5 & 6 in the revised manuscript. However, this sub-loop in the LRES domain does not interact with the upstream LREA region and therefore our data is consistent with our finding that the KLK locus is compartmentalised into active gene expression upstream of the persistent sites and repressive gene expression downstream of the persistent sites.

(3) Why is there still CTCF binding after CRISPR-Cas9 deletion of the motifs? Is it due to incomplete deletion? Or just persistent CTCF binding without the motif?

Response:

The CRISPR experiments were performed in a population of cells and therefore some cells retain their CTCF motif. See **Figure 5A** and methods. We were unable to isolate single CRISPR clones with both CTCF sites deleted even after multiple attempts, as the single cells were not viable.

(4) The error bars seem to be very large for LRES TAD boundary in Fig 4C. Since it is only duplicates for each data point, can the independent replicates values be shown instead of the error bar? Also why is the interaction frequency scale so different to Fig 1C?

Response:

We have now shown independent replicates. These are shown in **new Figure 5D and the new Supp. Fig 6A**. The interaction frequencies for 5D (previously 4D and 2A (previously 1B) are now on the same scales.

(5) If the goal is to show that the persistent sites have distinct chromatin characteristics, it would it be a fairer comparison to only compare persistent and lost CTCF peaks with similar CTCF levels for the NOME-seq and motif enrichment analysis. The lost peaks with weaker CTCF binding can be expected to lose CTCF binding first upon CTCF depletion in any case.

Response:

To address this concern we have now compared the persistent and lost CTCF peaks with high CTCF levels - we analysed the subset of lost CTCF sites with the highest CTCF levels (2973) to an equal number of persistent sites. We found that the CpG methylation levels are equivalent at the strongly bound CTCF sites and accessibility is also similar between the two groups. However peak height immediately over the CTCF binding site is more pronounced for persistent sites, which suggests that these sites are more consistently bound/occupied across a population of cells. We have now added this result to **new Supplementary Figure 8B** and results on page 11. We also performed the motif enrichment analysis on this ‘strongly bound’ subset of lost sites. This did increase the percentage of peaks with the binding motif identified (now

73%), when compared to the total lost subset (59.5%). However this was still not as high as the percentage of persistent sites with the motif (85.9%).

Reviewer #3 (Remarks to the Author):

In this manuscript, Khoury and colleagues describe the impact of CTCF decrease on gene expression and chromatin architecture. In line with work from others (Schmidt et al., 2012 Cell), knockdown of CTCF resulted in a selective loss of specific CTCF sites, while other regions were not affected. The ‘persistent’ CTCF sites were generally shared between different cell types, while the CTCF sites lost after siRNA were more cell type specific, again in line with Schmidt et al. Much to my surprise, this other manuscript is not cited or mentioned. As novel findings, it is now shown that depletion of CTCF resulted in alteration in the TAD structure. The ‘persistent’ CTCF sites were at TAD boundaries that were not altered after knockdown. Altogether, I find this an interesting manuscript, with novel interesting findings regarding the impact of CTCF on the 3D genome architecture. That said, novelty in relation to the 2012 Schmidt et al Cell paper is a matter of debate, as many of the claims (for example the entire Figure 7) have already been described by others.

Specific comments:

Page 7, line 158:

The authors state there are a subset of ‘persistent’ CTCF sites to siRNA CTCF. In 2012, Schmidt et al. Cell identified highly conserved CTCF sites across species, which were resistant to knockdown. As such, many of the conclusions drawn in the current work have also been reported by Schmidt et al. (practically the entire Figure 7), yet that paper is not cited in the current manuscript. The authors should include information of the overlap of their ‘persistent’ CTCF sites and these conserved sites (human), and place their findings into context with the data reported by Schmidt et al.

Response:

We have now included references to Schmidt et al (2010, 2012) in context to our new findings on page 15 in the discussion. However it is important to point out that Schmidt et al. present ChIP-seq data based on *only a ~50% CTCF knockdown* i.e. the control MCF7 cells had 73984 bound CTCF sites, while knockdown MCF7 cells had 38688 and they make the following statement ‘We analyzed CTCF binding before and after RNAi knockdown in human MCF-7 cells and found that virtually all binding events conserved across five species were resistant to knockdown, compared to only 60% of human-specific binding events’.

We believe that in our new study by increasing the efficiency of knockdown to 80-90% (and identified 2973/1840 persistent CTCF sites in LNCaP/IMR90 cells respectively), that we have more stringently identified a core set of conserved CTCF sites more resistant to CTCF knockdown. This is further supported by the remarkable overlap (87%) of these sites between 2 different cell types cancer (LNCaP) and normal (IMR90).

In addition to this, we also remove two of these core sites using CRISPR-Cas9n and demonstrate the consequences of their perturbation.

Page 7, line 162:

There is discussion of ‘persistent’ sites and ‘lost’ sites, but not of gained sites. Are there gained sites? And if so, what are the characteristics of these sites?
Reprogramming of CTCF in response to knockdown may be important biologically.

Response:

49 new CTCF sites (1.6%) were gained following CTCF knockdown versus 22644 sites lost (89%). We therefore did not investigate the characteristics of these sites as they constitute only a very small proportion (compared to thousands of CTCF sites retained following CTCF knockdown). We have now added this data to the **new Figure 4A** and the results on page 7.

Page 9, line 218:

A measurement of sequence conservation like PhastCons might also be revealing here between persistent and lost sites.

Response:

We have now used PhastCons to measure levels of evolutionary conservation and found that the persistent CTCF sites do in fact show greater sequence conservation when compared to the lost sites. This is concordant with our finding that persistent CTCF sites regulate constitutive chromatin architecture. We have included this as a new figure (**Supp. Figure 11B**) and added this to the results on page 13.

Page 14, line 335:

The claim is too strong that the alteration of TAD boundary structure occurs only at lost CTCF sites. Is there any statistical analysis to back this statement up?

Response:

We agree and have modified our statement to emphasise that persistent CTCF sites are enriched at stable TAD boundaries which we demonstrated statistically in **Figure 6E**.

Figure 4D: It’s not clear to me how the authors conclude a 66.1% reduction in chromatin interaction strength. The lines are mostly overlapping, with the exception of the site in the final gene, CTU1. What’s the basis of the 66,1% reduction conclusion?

Response:

We show that the bait fragment at the LREA/LRES boundary only makes one downstream interaction and this is with the site at CTU1 (see new Figure 2). The proximity of these two anchor points encloses the intervening chromatin into the downstream loop (as depicted in **new Figure 5E**, top panel). We found a 66.1% reduction in the chromatin interaction at the downstream CTU1 site based on the 3C-qPCR results averaged from 2 replicate experiments (**Figure 5D** and **new Supplementary Fig 6A**). As this is the only point of interaction, reduction in strength of this interaction is sufficient to weaken the downstream loop.

Hence in the paper we say ‘On average a 66.1% reduction in chromatin interaction strength was observed for the LRES loop’. We apologise if this was unclear and have changed the sentence to: ‘On average a 66.1% reduction in chromatin interaction strength was observed between the bait and its downstream interacting fragment’ on page 9.

Figure 5A: Include standard error shading in the profiles.

Response:

We generated the plots with error shading but found the error was too small to be visible (see below). The heat map in **Supp. Figure 7B** demonstrates the binding intensity level for all the lost sites and thus provides insight into the spread of intensities that are used to generate the average plots in **Supp. Figure 7A**.

Figure 5C: Zoom in is necessary to see differences. As far as I can tell, the methods for NOME-seq are not included. As this method relies on bisulfite conversion, a control sample needs to be shown as well.

Response:

We have enlarged the Figure and moved to **new Supplementary Figure 8**.

NOME-seq is a sensitive method to detect nucleosome positioning and strength of transcription factor binding. We used the NOME-seq data from our previous paper (Taberlay et al 2014) as referred to on page 10. NOME-seq relies on optimisation of the GpC methylase conditions to only methylated DNA in chromatin that is nucleosome/transcription factor free. The method including optimisation conditions and bisulphite treatment conditions are described in the Taberlay 2014 paper.

REVIEWERS' COMMENTS:

Reviewer #1 (Remarks to the Author):

The authors have properly addressed all technical questions raised in my review. Although I still have some doubts concerning the originality of the MS, it will be fair to give readers the opportunity to make their own judgement. Hence, I recommend this MS for publication.

Reviewer #2 (Remarks to the Author):

The authors have now addressed all my major concerns and the claims of the existence of persistent CTCF binding sites are now better supported by the data.

My only remaining query relates to the new Figure 5D and Supp. Fig 6A. I previously requested that the independent replicate values be shown rather than using error bars as these seem quite large. In the revision, the plots seems to be the same as before and the error bars are still there and the individual values for the replicates are still not shown. I am wondering if there is some misunderstanding?

Please find our point by point response to the REVIEWERS' COMMENTS:

Reviewer #1 (Remarks to the Author):

The authors have properly addressed all technical questions raised in my review. Although I still have some doubts concerning the originality of the MS, it will be fair to give readers the opportunity to make their own judgement. Hence, I recommend this MS for publication.

Response:

Thankyou

Reviewer #2 (Remarks to the Author):

The authors have now addressed all my major concerns and the claims of the existence of persistent CTCF binding sites are now better supported by the data.

My only remaining query relates to the new Figure 5D and Supp. Fig 6A. I previously requested that the independent replicate values be shown rather than using error bars as these seem quite large. In the revision, the plots seem to be the same as before and the error bars are still there and the individual values for the replicates are still not shown. I am wondering if there is some misunderstanding?

Response:

Apologies if there was any misunderstanding-

As requested, we have now included the graphs for each of the independent technical replicate qPCR values for both biological 3C replicate experiments (see revised Supp. Fig 6A and 6B).

We have also kept the Figures 5D and Supp. Fig. 6B for the biological 3C replicate experiments as we think that these provide a nice summary of the data.

We have now modified the text now in green for further clarification on page 9 and below and Figure legends (p39-40, p44).

“To determine if the gene activation was accompanied by a change in chromatin conformation, we performed 3C on the CTCF-CRISPR-Cas9n and non-targeting CRISPR-Cas9n control cells in biological duplicate experiments (Figure 5D and Supplementary Figure 6A, 6B). On average a 66.1% reduction in chromatin interaction strength was observed, in the 3C replicates, between the bait and its downstream interacting fragment for the LRES loop containing the up-regulated genes with a lesser change to the LREA loop (12.6% reduction) (Figure 5D and Supplementary Figure 6A, 6B).”

Please find the revised manuscript attached and revised Supp Fig. 6 which displays the independent technical replicate qPCR values for both biological 3C replicate experiments.

We hope that this is now all in order.

Best Wishes Sue